# PROVABLY ROBUST CLASSIFICATION OF ADVERSARIAL EXAMPLES WITH DETECTION

**Fatemeh Sheikholeslami**
Bosch Center for Artificial Intelligence
Pittsburgh, PA
`fatemeh.sheikholeslami@us.bosch.com`

**Ali Lotfi Rezaabad** *
The University of Texas at Austin
Austin, TX
`alotfi@utexas.edu`

**J. Zico Kolter**
Bosch Center for Artificial Intelligence
Carnegie Mellon University
Pittsburgh, PA
`zkolter@cs.cmu.edu`

## ABSTRACT

Adversarial attacks against deep networks can be defended against either by building robust classifiers or, by creating classifiers that can *detect* the presence of adversarial perturbations. Although it may intuitively seem easier to simply detect attacks rather than build a robust classifier, this has not bourne out in practice even empirically, as most detection methods have subsequently been broken by adaptive attacks, thus necessitating *verifiable* performance for detection mechanisms. In this paper, we propose a new method for jointly training a provably robust classifier and detector. Specifically, we show that by introducing an additional "abstain/detection" into a classifier, we can modify existing certified defense mechanisms to allow the classifier to either robustly classify *or* detect adversarial attacks. We extend the common interval bound propagation (IBP) method for certified robustness under $\ell_\infty$ perturbations to account for our new robust objective, and show that the method outperforms traditional IBP used in isolation, especially for large perturbation sizes. Specifically, tests on MNIST and CIFAR-10 datasets exhibit promising results, for example with provable robust error less than 63.63% and 67.92%, for 55.6% and 66.37% natural error, for $\epsilon = 8/255$ and $16/255$ on the CIFAR-10 dataset, respectively.

## 1 INTRODUCTION

Despite popularity and success of deep neural networks in many applications, their performance declines sharply in adversarial settings. Small adversarial perturbations are shown to greatly deteriorate the performance of neural network classifiers, which creates a growing concern for utilizing them in safety critical application where robust performance is key. In adversarial training, different methods with varying levels of computational complexity aim at robustifying the network by finding such *adversarial examples* at each training steps and adding them to the training dataset. While such methods exhibit empirical robustness, they lack verifiable guarantees as it is not provable that a more rigorous adversary, e.g., one that does brute-force enumeration to compute adversarial perturbations, will not be able to cause the classifier to misclassify.

It is thus desirable to provably verify the performance of robust classifiers without restricting the adversarial perturbations by inexact solvers, while restraining perturbations to a class of admissible set, e.g., within an $\ell_\infty$ norm-bounded ball. Progress has been made by 'complete methods' that use Satisfiability Modulo Theory (SMT) or Mixed-Integer Programming (MIP) to provide exact robustness bounds, however, such approaches are expensive, and difficult to scale to large networks as exhaustive enumeration in the worst case is required (Tjeng et al., 2017; Ehlers, 2017; Xiao et al., 2018).

---

*Work was done when the author was an intern at Bosch Center for Artificial Intelligence, Pittsburgh, PA.

'Incomplete methods' on the other hand, proceed by computing a differential upper bound on the worst-case adversarial loss, and similarly for the verification violations, with lower computational complexity and improved scalability. Such upper bounds, if easy to compute, can be utilized during the training, and yield provably robust networks with tight bounds. In particular, bound propagation via various methods such as differentiable geometric abstractions (Mirman et al., 2018), convex polytope relaxation (Wong & Kolter, 2018), and more recently in (Salman et al., 2019; Balunovic & Vechev, 2020; Gowal et al., 2018; Zhang et al., 2020), together with other techniques such as semi-definite relaxation, (Fazlyab et al., 2019; Raghunathan et al., 2018), and dual solutions via additional verifier networks (Dvijotham et al., 2018) fall within this category. In particular, recent successful use of Interval Bound Propagation (IBP) as a simple layer-by-layer bound propagation mechanism was shown to be very effective in Gowal et al. (2018), which despite its light computational complexity exhibits SOTA robustness verification. Additionally, combining IBP in a forward bounding pass with linear relaxation based backward bounding pass (CROWN) Zhang et al. (2020) leads to improved robustness, although it can be up to 3-10 times slower.

Alternative to robust classification, detection of adversarial examples can also provide robustness against adversarial attacks, where suspicious inputs will be flagged and the classifier "rejects/abstains" from assigning a label. There has been some work on detection of out-of-distribution examples Bitterwolf et al. (2020), however the situation in the literature on the detection of adversarial examples is quite different from above. Most techniques that attempt to detect adversarial examples, either by training explicit classifiers to do so or by simply formulating "hand-tuned" detectors, still largely look to identify and exploit statistical properties of adversarial examples that appear in practice Smith & Gal (2018); Roth et al. (2019). However, to provide a fair evaluation, a defense must be evaluated under attackers that attempt to fool both the classifier *and* the detector, while addressing particular characteristics of a given defense, e.g., gradient obfuscation, non-differentability, randomization, and simplifying the attacker's objective for increased efficiency. A non-exhaustive list of recent detection methods entails randomization and sparsity-based defenses (Xiao et al., 2019; Roth et al., 2019; Pang et al., 2019b), confidence and uncertainty-based detection (Smith & Gal, 2018; Stutz et al., 2020; Sheikholeslami et al., 2020), transformation-based defenses (Bafna et al., 2018; Yang et al., 2019), ensemble methods (Verma & Swami, 2019; Pang et al., 2019a), generative adversarial training Yin et al. (2020), and many more. Unfortunately, existing defenses have largely proven to have poor performance against adaptive attacks (Athalye et al., 2018; Tramer et al., 2020), necessitating provable guarantees on detectors as well. Recently Laidlaw & Feizi (2019) have proposed joint training of classifier and detector, however it also does not provided any provable guarantees.

**Our contribution**. In this work, we propose a new method for jointly training a provably robust classifier and detector. Specifically, by introducing an additional "abstain/detection" into a classifier, we show that the existing certified defense mechanisms can be modified, and by building on the detection capability of the network, classifier can effectively *choose* to either robustly classify *or* detect adversarial attacks. We extend the light-weight Interval Bound Propagation (IBP) method to account for our new robust objective, enabling verification of the network for provable performance guarantees. Our proposed robust training objective is also effectively upper bounded, enabling its incorporation into the training procedure leading to tight provably robust performance. While tightening of the bound propagation may be additionally possible for tighter verification, to the best of our knowledge, our approach is the first method to extend certification techniques by considering *detection* while providing *provable* verification. By stabilizing the training, as also used in similar IBP-based methods, experiments on MNIST and CIFAR-10 empirically show that the proposed method can successfully leverage its detection capability, and improves traditional IBP used in isolation, especially for large perturbation sizes.

## 2 BACKGROUND AND RELATED WORK

Let us consider an $L$-layer feed-forward neural network, trained for a $K$-class classification task. Given input $\mathbf{x}$, it will pass through a sequential model, with $h_l$ denoting the mapping at layer $l$, recursively parameterized by

$$\mathbf{z}_l = h_l(\mathbf{z}_{l-1}) = \sigma_l(\mathbf{W}_l^\top \mathbf{z}_{l-1} + \mathbf{b}_l),\ l = 1, \cdots, L \quad \mathbf{W}_l \in \mathbb{R}^{n_{l-1} \times n_l}, \quad \mathbf{b}_l \in \mathbb{R}^{n_l} \qquad (1)$$

where $\sigma_l(.)$ is a monotonic activation function, $\mathbf{z}_0$ denotes the input, and $\mathbf{z}_L \in \mathbb{R}^K$ is the pre-activation unnormalized $K$-dimensional output vector ($n_L = K$ and $\sigma_L(.)$ as identity operator),

referred to as the logits. Robust classifiers can be obtained by minimizing the worst-case (adversarial) classification loss, formally trained by the following min-max optimization Madry et al. (2017)

$$\underset{\theta}{\text{minimize}} \quad \underset{(\mathbf{x},y)\sim\mathcal{D}}{\mathbb{E}} \left[ \max_{\boldsymbol{\delta}\in\Delta_\epsilon} \ell(f_\theta(\mathbf{x}+\boldsymbol{\delta}),y) \right]. \tag{2}$$

where $\theta$ denotes network parameters, vector $f_\theta(\mathbf{x}) = \mathbf{z}_L$ is the logit output for input $\mathbf{x}$, $\ell(.)$ is the misclassification loss, e.g., $\ell_{\text{xent}}(.)$ defined as the cross-entropy loss, and $\Delta_\epsilon$ denotes the set of permissible perturbations, e.g., for $\ell_\infty$-norm ball of radius $\epsilon$ giving $\Delta_\epsilon := \{\boldsymbol{\delta} \mid \|\boldsymbol{\delta}\|_\infty \le \epsilon\}$. Although augmenting the training set with adversarial inputs, obtained by approximately solving the inner maximization in Eq. 2, empirically leads to improved adversarial robustness Madry et al. (2017); Shafahi et al. (2019); Wong et al. (2019); Zhang et al. (2019), inexact solution of the inner maximization prevents such methods from providing provable guarantees. In critical applications however, provable verification of classification accuracy against a given threat model is crucial.

## 2.1 PERFORMANCE VERIFICATION AND NETWORK RELAXATION

Given input $(\mathbf{x}, y)$, a classification network is considered verifiably robust if all of its perturbed variations, that is $\mathbf{x} + \boldsymbol{\delta}$ for $\forall\boldsymbol{\delta} \in \Delta_\epsilon$, are correctly classified as class $y$. Such verification can be effectively obtained by

$$p_i^* = \min_{\mathbf{z}_L\in\mathcal{Z}_L} \mathbf{c}_{y,i}^\top \mathbf{z}_L \quad \text{where}, \quad \mathcal{Z}_L := \{\mathbf{z}_L|\mathbf{z}_l = h_l(\mathbf{z}_{l-1}),\ l=1,...,L,\ \mathbf{z}_0 = \mathbf{x} + \boldsymbol{\delta},\ \forall\boldsymbol{\delta}\in\Delta_\epsilon\}$$

where $\mathbf{c}_{y,i} = \mathbf{e}_y - \mathbf{e}_i$ for $i = 1, 2, .., K, i \ne y$, and $\mathbf{e}_i$ is the standard $i^{\text{th}}$ canonical basis vector. If $p_i^* > 0$ $\forall i \ne y$, then the classifier is verifiably robust at point $(\mathbf{x}, y)$ as this guarantees that $z_i \le z_y$ $\forall i \ne y$ for all admissible perturbations $\boldsymbol{\delta} \in \Delta_\epsilon$.

The feasible set $\mathcal{Z}_L$ is generally nonconvex, rendering obtaining $p_i^*$ intractable. Any convex relaxation of $\mathcal{Z}_L$ however, will provide a lower bound on $p_i^*$, and can be alternatively used for verification. As outlined in Section 1, various relaxation techniques have been proposed in the literature. Specifically, IBP in (Mirman et al., 2018; Gowal et al., 2018) proceeds by bounding the activation $\mathbf{z}_l$ of each layer by propagating an element-wise bounding box using interval arithmetic for networks with monotonic activation functions. Despite its simplicity and relatively small computational complexity (computational requirements for bound propagation for a given input using IBP is equal to two forward passes of the input), it can provide tight bounds once the network is trained accordingly.

Specifically, starting from the input layer $\mathbf{z}_0$, it can be bounded for the perturbation class $\boldsymbol{\delta} \in \Delta_\epsilon$ as $\underline{\mathbf{z}}_0 = \mathbf{x} - \epsilon\mathbf{1}$ and $\overline{\mathbf{z}}_0 = \mathbf{x} + \epsilon\mathbf{1}$, and $\mathbf{z}_l$ for the following layers can be bounded as

$$\underline{\mathbf{z}}_l = \sigma_l(\mathbf{W}_l^\top \frac{\overline{\mathbf{z}}_{l-1} + \underline{\mathbf{z}}_{l-1}}{2} - |\mathbf{W}_l^\top| \frac{\overline{\mathbf{z}}_{l-1} - \underline{\mathbf{z}}_{l-1}}{2}), \quad \overline{\mathbf{z}}_l = \sigma_l(\mathbf{W}_l^\top \frac{\overline{\mathbf{z}}_{l-1} + \underline{\mathbf{z}}_{l-1}}{2} + |\mathbf{W}_l^\top| \frac{\overline{\mathbf{z}}_{l-1} - \underline{\mathbf{z}}_{l-1}}{2}), \tag{3}$$

where $|\cdot|$ is the element-wise absolute-value operator. The verification problem over the relaxed feasible set $\hat{\mathcal{Z}}_L := \{\mathbf{z}_L \mid \underline{z}_{L,i} \le z_{L,i} \le \bar{z}_{L,i}\}$, where $\mathcal{Z}_L \subseteq \hat{\mathcal{Z}}_L$ is then easily solved as

$$p_i^* = \min_{\mathbf{z}_L\in\mathcal{Z}_L} \mathbf{c}_{y,i}^\top \mathbf{z}_L \ge \min_{\mathbf{z}_L\in\hat{\mathcal{Z}}_L} \mathbf{c}_{y,i}^\top \mathbf{z}_L = \underline{z}_{L,y} - \bar{z}_{L,i}. \tag{4}$$

## 2.2 ROBUST TRAINING OF VERIFIABLE NETWORKS

It has been shown that convex relaxation of $\mathcal{Z}_L$ can also provide a tractable upper bound on the inner maximization in Eq. 2. While this holds for various relaxation techniques, focusing on the IBP let us define

$$\mathbf{J}_{\epsilon,\theta}^{\text{IBP}}(\mathbf{x},y) := [J_1^{\text{IBP}}, J_2^{\text{IBP}}, ..., J_K^{\text{IBP}}] \quad \text{where} \quad J_i^{\text{IBP}} := \min_{\mathbf{z}_L\in\hat{\mathcal{Z}}_L} \mathbf{c}_{y,i}^\top \mathbf{z}_L \tag{5}$$

with $(\theta, \epsilon)$ implicitly influencing $\hat{\mathcal{Z}}_L$ (dropped for brevity), and upperbound the inner-max in Eq. 2

$$\max_{\boldsymbol{\delta}\in\Delta_\epsilon} \ell_{\text{xent}}(f_\theta(\mathbf{x}+\boldsymbol{\delta}),y) \le \ell_{\text{xent}}(-\mathbf{J}_{\epsilon,\theta}^{\text{IBP}}(\mathbf{x},y),y), \quad \ell_{\text{xent}}(\mathbf{z},c) := -\log\left(\frac{\exp(z_c)}{\sum_i \exp(z_i)}\right) \tag{6}$$

By using this tractable upper bound of the robust optimization, network can now be trained by

$$\underset{\theta}{\text{minimize}} \quad \sum_{(\mathbf{x},y)\in\mathcal{D}} (1-\kappa)\ell_{\text{xent}}(-\mathbf{J}_{\epsilon,\theta}^{\text{IBP}}(\mathbf{x},y),y) + \kappa\gamma\ell_{\text{xent}}(f_\theta(\mathbf{x}),y), \tag{7}$$

where $\gamma$ trades natural versus robust accuracy, and $\kappa$ is scheduled through a ramp-down process to stabilize the training and tightening of IBP Gowal et al. (2018) (where $\gamma = 1$ is selected therein).

# 3 VERIFIABLE CLASSIFICATION WITH DETECTION

In this paper, we propose a new method for jointly training a provably robust classifier and detector. Specifically, let us augment the classifier by introducing an additional "abstain/detection". This can be readily done by extending the $K$-class classification task to a $(K + 1)$-class classification, with the $(K + 1)$-th class dedicated to the detection task, and the maximum weighted class is finally chosen as the classification output. The classifier is then trained such that adversarial examples, or *ideally any other example that the network would misclassify*, are classified in this abstain class, denoted by $a$, thus preventing incorrect classification.

Formally, the classifier can be denoted as in Eq. 1, with the only difference that the final output is $K + 1$ dimensional, i.e., $\mathbf{z}_L \in \mathbb{R}^{K+1}$; simply by substituting the last fully-connected weight matrix $\mathbf{W}_L$ of dimension $n_L \times K$ with that of dimension $n_L \times (K + 1)$, and similarly for $\mathbf{b}_L$.

## 3.1 VERIFICATION PROBLEM FOR CLASSIFICATION WITH ABSTAIN/DETECTION

It is desirable to provably verify performance of the joint classification/detection. In contrast to existing robust classifiers, however, on a perturbed image $\mathbf{x} + \boldsymbol{\delta}$, the classification/detection task is considered successful if the input is classified either as the correct class $y$, *or* as the abstain class $a$; as both cases prevent misclassification of the adversarially perturbed input as a wrong class. On clean natural images however, classification/detection is considered successful *only* if it is classified as the correct class $y$, and abstaining is considered misclassification.

In order to certify performance in adversarial settings, it is now sufficient to verify that the network satisfies the following for a given input pair $(\mathbf{x}, y)$ and $\boldsymbol{\delta} \in \Delta_\epsilon$ and $i = 1, .., K, i \neq y$:

$$\max\{\mathbf{c}_{y,i}^\top \mathbf{z}, \mathbf{c}_{a,i}^\top \mathbf{z}\} \geq 0 \qquad \forall \mathbf{z} \in \mathcal{Z}_L := \{\mathbf{z}_L | \mathbf{z}_l = h_l(\mathbf{z}_{l-1}) \; l = 1, ..., L, \; \mathbf{z_0} = \mathbf{x} + \boldsymbol{\delta}, \; \forall \boldsymbol{\delta} \in \Delta_\epsilon\} \tag{8}$$

where $\mathbf{c}_y := \mathbf{e}_y - \mathbf{e}_i$ and $\mathbf{c}_a := \mathbf{e}_a - \mathbf{e}_i$, $a$ denotes the "abstain" class, and the dependence of $\mathcal{Z}_L$ on $(\mathbf{x}, y, \epsilon, \theta)$ is omitted for brevity. Verification can be done effectively by seeking a counterexample

$$\pi_i^* := \min_{\mathbf{z} \in \mathcal{Z}_L} \quad \max\{\mathbf{c}_{y,i}^\top \mathbf{z}, \mathbf{c}_{a,i}^\top \mathbf{z}\}. \tag{9}$$

If $\pi_i^* \geq 0 \; \forall i \neq y$ , the specification is then satisfied and the performance is verified.

Similar to previous verification methods, to overcome the non-convexity of the optimization in Eq. 9, one can lower bound the problem by expanding the feasible set $\mathcal{Z}_L \subseteq \hat{\mathcal{Z}}_L$ , where $\hat{\mathcal{Z}}_L$ is convex, as stated in Theorem 1, and proved in Appendix A.1.

**Theorem 1:** *For any convex $\hat{\mathcal{Z}}_L$ s.t. $\mathcal{Z}_L \subseteq \hat{\mathcal{Z}}_L$, Eq. 9 can be bounded by the convex relaxation*

$$\max_{0 \leq \eta \leq 1} \min_{\mathbf{z} \in \hat{\mathcal{Z}}_L} \left(\eta\, \mathbf{c}_{a,i} + (1 - \eta)\, \mathbf{c}_{y,i}\right)^\top \mathbf{z} \leq \min_{\mathbf{z} \in \mathcal{Z}_L} \max\{\mathbf{c}_{y,i}^\top \mathbf{z}, \mathbf{c}_{a,i}^\top \mathbf{z}\}. \tag{10}$$

Although Theorem 1 holds for any convex relaxation of $\mathcal{Z}_L$, for IBP relaxation in Gowal et al. (2018) it can be further simplified by substituting $\mathbf{z} = \mathbf{W}_L^\top \mathbf{z}_{L-1} + \mathbf{b}_L$, thus not propagating the intervals through the last layer for tighter bounding, and solved analytically as follows.

**Theorem 2:** *The optimization in Eq. 9 can be lower-bounded by the convex optimization*

$$J_i(\mathbf{x}, y) = \max_{0 \leq \eta \leq 1} \min_{\mathbf{z}_{L-1} \in \hat{\mathcal{Z}}_{L-1}} (\boldsymbol{\omega}_1 + \eta\, \boldsymbol{\omega}_2)^\top \mathbf{z}_{L-1} + \eta\, \omega_3 + \omega_4 \quad \leq \min_{\mathbf{z} \in \mathcal{Z}_L} \max\{\mathbf{c}_{y,i}^\top \mathbf{z}, \mathbf{c}_{a,i}^\top \mathbf{z}\} \tag{11}$$

*in which $\boldsymbol{\omega}_1 := \mathbf{W}_L^\top \mathbf{c}_{y,i}$, $\boldsymbol{\omega}_2 := \mathbf{W}_L^\top (\mathbf{c}_{a,i} - \mathbf{c}_{y,i})$, $\omega_3 := \mathbf{b}_L^\top (\mathbf{c}_{a,i} - \mathbf{c}_{y,i})$, $\omega_4 := \mathbf{b}_L^\top \mathbf{c}_{y,i}$ and convex set $\hat{\mathcal{Z}}_{L-1}$ is a convex relaxation of $\mathcal{Z}_{L-1}$ on the hidden values at $L - 1$. Furthermore, $J_i(\mathbf{x}, y)$ can be analytically obtained as outlined in Alg. 1*

Note that since $\eta$ is the dual variable, any selection within the feasible set serves as a (looser but valid) lower bound, while the maximization makes the bound tight; see Appendix A.2 and A.3 for proof and a step-by-step algorithm description. Similar to other convex relaxation-based verification methods, in order for a networks to provide verifiable performance, one needs to incorporate bound propagation in training.

---

**Algorithm 1** Solution for $J_i(\mathbf{x}, y)$ in Theorem 2

1: **Input.** Bounds on layer $L - 1 : \underline{\mathbf{z}}_{L-1}, \bar{\mathbf{z}}_{L-1}$, and weight matrix $\mathbf{W}_L$
2: $\boldsymbol{\omega}_1 = \mathbf{W}_L \mathbf{c}_{y,i}$ and $\boldsymbol{\omega}_2 = \mathbf{W}_L(\mathbf{c}_{a,i} - \mathbf{c}_{y,i})$, $\omega_3 := \mathbf{b}_L^\top(\mathbf{c}_{a,i} - \mathbf{c}_{y,i})$, and $\omega_4 := \mathbf{b}_L^\top \mathbf{c}_{y,i}$
3: $\boldsymbol{\zeta} = [\zeta_1, ..., \zeta_{n_L}] := -\boldsymbol{\omega}_1/\boldsymbol{\omega}_2$ and vector of indices $\mathbf{s}$ that sorts $\boldsymbol{\zeta}$ , i.e., $\zeta_{s_1} \leq \cdots \leq \zeta_{s_{n_{L-1}}}$
4: $\underline{\mathbf{u}}_1 = \Pi_{\mathbf{s}}(\boldsymbol{\omega}_1 \circ \underline{\mathbf{z}}_{L-1})$, $\bar{\mathbf{u}}_1 = \Pi_{\mathbf{s}}(\boldsymbol{\omega}_1 \circ \bar{\mathbf{z}}_{L-1})$, $\underline{\mathbf{u}}_2 := \Pi_{\mathbf{s}}(\boldsymbol{\omega}_2 \circ \underline{\mathbf{z}}_{L-1})$, $\bar{\mathbf{u}}_2 := \Pi_{\mathbf{s}}(\boldsymbol{\omega}_2 \circ \bar{\mathbf{z}}_{L-1})$
   where operators $\circ$ and $\Pi_{\mathbf{s}}(.)$ denote element-wise multiplication, and permutation according to indices $\mathbf{s}$, respectively.
5: $m = \min_{\zeta_{s_j} \geq 0} j$ and $M = \max_{\zeta_{s_j} \leq 1} j$ for $j = 1, ..., n_{L-1}$
6: **for** $\eta = 0, \zeta_{s_m}, \zeta_{s_{m+1}}, \cdots, \zeta_{s_{M-1}}, \zeta_{s_M}, 1$ **do**
7:     Compute

$$g(\eta) = \sum_{j=1}^{n_{L-1}} \left( 1_{\{\omega_{1,j} + \eta\omega_{2,j} \leq 0\}} \left(\bar{u}_{1,j} + \eta\bar{u}_{2,j}\right) + 1_{\{\omega_{1,j} + \eta\omega_{2,j} \geq 0\}} \left(\underline{u}_{1,j} + \eta\underline{u}_{2,j}\right) \right) + \eta \, \omega_3 + \omega_4$$

8: **return** $\max g(\eta)$ over the computed values.

---

## 4 TRAINING A VERIFIABLE ROBUST CLASSIFICATION WITH DETECTION

In order to train a robust classifier with detection, let us start by formalizing the objective of an adversarial attacker. Naturally, an adaptive attacker's objective is to craft perturbation $\boldsymbol{\delta}$ such that it simultaneously evades detection and causes misclassification. Formally, this can be tackled by seeking $\boldsymbol{\delta}$ such that loss corresponding to the winner of the two classes $y$ and $a$ (higher logit leading to smaller cross-entropy loss) is maximized, i.e.,

$$\max_{\boldsymbol{\delta} \in \Delta} \min \left\{ \ell_{\text{xent}}(f_\theta(\mathbf{x} + \boldsymbol{\delta}), y), \ell_{\text{xent}}(f_\theta(\mathbf{x} + \boldsymbol{\delta}), a) \right\} \tag{12}$$

where $\ell_{\text{xent}}(\mathbf{z}, c)$ denotes the cross-entropy loss for class $c = y$ and $c = a$, and $\mathcal{I} = \{1, 2, ..., K, a\}$ denotes the class index set with $K + 1$ elements.

Let us now define

$$L_{\text{robust}}^{\text{abstain}}(\mathbf{x}, y; \theta) := \max_{\boldsymbol{\delta} \in \Delta} \min \left\{ \ell_{\text{xent}\backslash a}(f_\theta(\mathbf{x} + \boldsymbol{\delta}), y), \ell_{\text{xent}\backslash y}(f_\theta(\mathbf{x} + \boldsymbol{\delta}), a) \right\} \tag{13}$$

in which the inner maximization is closely related to that of the adversarial objective in Eq. 12 with a small difference: loss terms $\ell_{\text{xent}\backslash a}$ and $\ell_{\text{xent}\backslash y}$ are defined as

$$\ell_{\text{xent}\backslash a}(\mathbf{z}, y) := -\log\left(\frac{\exp(z_y)}{\sum_{i \in \mathcal{I}\backslash\{a\}} \exp(z_i)}\right), \text{ and } \ell_{\text{xent}\backslash y}(\mathbf{x}, a) := -\log\left(\frac{\exp(z_a)}{\sum_{i \in \mathcal{I}\backslash\{y\}} \exp(z_i)}\right).$$

This small alteration to the cost, while not changing the minimization "winner" between the true class $y$ and rejection class $a$ in Eq. 12 and 13, i.e.,

$$\begin{cases} z_a \leq z_y \Rightarrow & \ell_{\text{xent}}(f_\theta(\mathbf{x} + \boldsymbol{\delta}), y) \leq \ell_{\text{xent}}(f_\theta(\mathbf{x} + \boldsymbol{\delta}), a) \text{ and } \ell_{\text{xent}\backslash a}(f_\theta(\mathbf{x} + \boldsymbol{\delta}), y) \leq \ell_{\text{xent}\backslash y}(f_\theta(\mathbf{x} + \boldsymbol{\delta}), a) \\ z_y \leq z_a \Rightarrow & \ell_{\text{xent}}(f_\theta(\mathbf{x} + \boldsymbol{\delta}), a) \leq \ell_{\text{xent}}(f_\theta(\mathbf{x} + \boldsymbol{\delta}), y) \text{ and } \ell_{\text{xent}\backslash y}(f_\theta(\mathbf{x} + \boldsymbol{\delta}), a) \leq \ell_{\text{xent}\backslash a}(f_\theta(\mathbf{x} + \boldsymbol{\delta}), y) \end{cases}$$

favorably influences the training process. That is so since, for $\boldsymbol{\delta}$ such that, for instance $z_a < z_y$, minimizing $L_{\text{robust}}^{\text{abstain}}(\mathbf{x}, y; \theta)$ during training reduces to minimizing $\ell_{\text{xent}}(f_\theta(\mathbf{x} + \boldsymbol{\delta}), y)$ which in turn leads to further increasing $z_y$ while *decreasing* the logit value $z_a$; and similarly, increasing $z_y$ while *decreasing* $z_y$ if $z_y < z_a$. Intuitively however, the true objective of the *classifier augmented with detection* on adversarial examples is to increase both $z_y$ and $z_a$ while reducing $z_j$, $\forall j \neq a, y$; thus preventing any *gap* in between the boundary of the classes $a$ and $y$, which can potentially lead to successful *adaptive* attacks. Hence, minimizing Eq. 12 would be in contrast with the true underlying objective, and Eq. 13 simply prevents the raised issue.

Upon defining $L_{\text{natural}}(\mathbf{x}, y; \theta) := \ell_{\text{xent}}(f_\theta(\mathbf{x}), y)$ and $L_{\text{robust}}(\mathbf{x}, y; \theta) := \max_{\boldsymbol{\delta} \in \Delta} \ell_{\text{xent}}(f_\theta(\mathbf{x} + \boldsymbol{\delta}), y)$, we then define the overall training loss as

$$L = L_{\text{robust}}(\mathbf{x}, y; \theta) + \lambda_1 L_{\text{robust}}^{\text{abstain}}(\mathbf{x}, y; \theta) + \lambda_2 L_{\text{natural}}(\mathbf{x}, y; \theta), \tag{14}$$

where $L_{\text{natural}}(\mathbf{x}, y; \theta)$ captures the misclassification loss of the natural (clean) examples, and $L_{\text{robust}}(\mathbf{x}, y; \theta)$ denotes that of adversarial examples without considering the rejection class, i.e., similar to that of Gowal et al. (2018), and parameters $(\lambda_1, \lambda_2)$ trade-off clean and adversarial accuracy. To train a robust classifier, we proceed by minimizing the overall loss Eq. 14, by first upperbounding $L_{\text{robust}}(\mathbf{x}, y; \theta)$ and $L_{\text{robust}}^{\text{abstain}}(\mathbf{x}, y; \theta)$.

## 4.1 UPPERBOUNDING THE TRAINING LOSS

Using Theorem 2, and restricting $0 < \underline{\eta} \leq \eta \leq \bar{\eta} < 1$, let us now define $J_i^{\eta, \bar{\eta}}(\mathbf{x}, y)$, where trivially

$$J_i^{\underline{\eta}, \bar{\eta}}(\mathbf{x}, y) := \max_{0 \leq \underline{\eta} \leq \eta \leq \bar{\eta} \leq 1} \quad (\boldsymbol{\omega}_1 + \eta \boldsymbol{\omega}_2)^\top \hat{\mathbf{z}}_{L-1} + \eta\, \omega_3 + \omega_4 \leq J_i(\mathbf{x}, y) \tag{15}$$

and can also be solved analytically similar to Theorem 2. By generalizing the findings in Wong & Kolter (2018); Mirman et al. (2018), we can upper bound the robust optimization problem using our dual problem in Eq. 15, according to the following Theorem, which we prove in Appendix A.4.

**Theorem 3:** *For any data point* $(\mathbf{x}, y)$*, and* $\epsilon > 0$*, and for any* $0 \leq \underline{\eta} \leq \bar{\eta} \leq 1$*, the adversarial loss* $L_{\text{robust}}^{\text{abstain}}(\mathbf{x}, y; \theta)$ *in Eq. 13 can be upper bounded by*

$$L_{\text{robust}}^{\text{abstain}}(\mathbf{x}, y; \theta) \leq \bar{L}_{\text{robust}}^{\text{abstain}}(\mathbf{x}, y; \theta) := \ell_{\text{xent} \backslash a}(-\mathbf{J}_{\epsilon, \theta}(\mathbf{x}, y), y) = \ell_{\text{xent} \backslash y}(-\mathbf{J}_{\epsilon, \theta}(\mathbf{x}, y), a) \tag{16}$$

*where* $\mathbf{J}_{\epsilon, \theta}(\mathbf{x}, y)$ *is a* $(K + 1)$*-dimensional vector, valued at index* $i$ *as* $[\mathbf{J}_{\epsilon, \theta}(\mathbf{x}, y)]_i = J_i^{\underline{\eta}, \bar{\eta}}(\mathbf{x}, y)$.

Note that maximization over $\eta$ for obtaining $J_i^{\underline{\eta}, \bar{\eta}}(\mathbf{x}, y)$ can be done either by bisection (concave maximization) or by following Alg. 1 and substituting $m = \min_{\zeta_{s_\nu} \geq \underline{\eta}} \nu$ , and $M = \max_{\zeta_{s_\nu} \leq \bar{\eta}} \nu$

**Remark 1**. Setting $\underline{\eta} = \bar{\eta} = 0$ forces $\eta = 0$ which reduces $J_i^{\underline{\eta}, \bar{\eta}}(\mathbf{x}, y)$ in Eq. 15 to that in Eq. 5, .i.e, $J_i^{\text{IBP}}(\mathbf{x}, y) = J_i^{\underline{\eta}, \bar{\eta}}(\mathbf{x}, y)|_{\underline{\eta} = \bar{\eta} = 0}$, also bounding loss term $L_{\text{natural}}(\mathbf{x}, y; \theta)$ as

$$L_{\text{robust}}(\mathbf{x}, y; \theta) \leq \bar{L}_{\text{robust}}(\mathbf{x}, y; \theta) := \ell_{\text{xent}}(-\mathbf{J}_{\epsilon, \theta}^{\text{IBP}}(\mathbf{x}, y), y). \tag{17}$$

**Remark 2**. While setting $\underline{\eta} = 0$ and $\bar{\eta} = 1$ gives tighter bounds, (and is thus used for the verification counterpart in Theorem 2), strictly setting $0 < \underline{\eta} \leq \bar{\eta} < 1$ empirically yields better generalization of the network. This can be intuitively understood by rewriting $\boldsymbol{\omega}_1 + \eta \boldsymbol{\omega}_2 = \mathbf{W}_L^\top (\eta \mathbf{c}_{a,i} + (1 - \eta) \mathbf{c}_{y,i})$ which is a convex combination of the verification constraints for the correct and the abstain class. Thus $\eta \neq 0 \neq 1$ will lead to minimizing a combination of both terms, preventing gaps in between the two classes. Also, higher values of $\eta$ increase the influence of the term corresponding to the abstain case, and vice versa, whose tuning can promote abstaining by considering how desirable such outcome is (or is not).

Utilizing upperbounds in Eq. 16 and Eq. 17, we can proceed to training the network by minimizing the tractable upperbound on the overall loss

$$\min_\theta L \leq \min_\theta \ell_{\text{xent}}(-\mathbf{J}_{\epsilon, \theta}^{\text{IBP}}(\mathbf{x}, y), y) + \lambda_1 \ell_{\text{xent} \backslash y}(-\mathbf{J}_{\epsilon, \theta}(\mathbf{x}, y), a) + \lambda_2 \ell_{\text{natural}}(f_\theta(\mathbf{x}), y) \tag{18}$$

Note that setting $\lambda_1 = 0$ and $\gamma = \lambda_2$ - and incorporation of a ramp-down process by parameter $\kappa$ as detailed in Section 5 - reduces the training in Eq. 18 to that of Gowal et al. (2018) without detection.

**Complexity.** Since given IBP bounds on $\mathbf{z}_{L-1}$, the solution to Eq. 16 is analytically available (that is after sorting whose complexity is negligible in comparison with forward pass), computing Eq. 18 imposes the same computational complexity as in IBP, which is twice the normal training procedure, as it requires propagating the upper and lower bounds via forward pass.

## 5 EXPERIMENTS

Empirical performance of the proposed robust classification with detection on MNIST-10 and CIFAR-10 datasets is reported in this section, and is compared with the state-of-the-art alternatives. The training procedure is stabilized as detailed next [1].

---

[1]Code is available at https://github.com/boschresearch/robust_classification_with_detection

## 5.1 STABILIZING THE TRAINING PROCEDURE

We incorporate the following mechanisms to stabilize the training procedure in our tests, where the first two have been previously used in (Gowal et al., 2018) and (Zhang et al., 2020) as well.

**Ramp down of $\kappa$:** To stabilize the trade-off between nominal and verified accuracy, let us introduce parameter $\kappa$ in the overall loss by trading the natural and robust loss as

$$L = (1 - \kappa) \underbrace{\left( \bar{L}_{\text{robust}}(\mathbf{x}, y; \theta) + \lambda_1 \bar{L}_{\text{robust}}^{\text{abstain}}(\mathbf{x}, y; \theta) \right)}_{\text{Robust loss}} + \kappa \underbrace{\lambda_2 L_{\text{natural}}(\mathbf{x}, y; \theta)}_{\text{Natural loss}} \tag{19}$$

Setting $\kappa = 0.5$ renders the optimization identical to that in Eq. 18. During the training however, we incorporate a ramp down procedure where $\kappa$ starts at value $\kappa_{\text{start}} = 1$, thus training the model to fit the natural data, and slowly decreasing it to value $\kappa_{\text{end}} = 0.5$, similar to that in Gowal et al. (2018).

**Ramp up of $\epsilon$:** It is very important during the training process to start at $\epsilon = 0$ and gradually increase it to $\epsilon_{\text{train}}$, while also setting $\epsilon_{\text{train}}$ larger than $\epsilon_{\text{test}}$ can improve generalization.

**Ramp down of $\underline{\eta}$ and $\bar{\eta}$:** Setting $0 < \underline{\eta}$ and $\bar{\eta} < 1$ helps with better generalization. Furthermore, setting large $\underline{\eta}$ and $\bar{\eta}$ promotes the abstain class in loss term $\bar{L}_{\text{robust}}^{\text{abstain}}$ by increasing the weight of $\boldsymbol{\omega}_2$ in Eq. 15. Thus, we can further stabilize the training process through a ramp down procedure where these parameters start at $\underline{\eta} = \underline{\eta}_{\text{start}}$ and $\bar{\eta} = \bar{\eta}_{\text{start}}$, and are gradually reduced to $\underline{\eta} = \underline{\eta}_{\text{end}}$ and $\bar{\eta} = \bar{\eta}_{\text{end}}$, with $\underline{\eta}_{\text{end}} < \underline{\eta}_{\text{start}}$ and $\bar{\eta}_{\text{end}} < \bar{\eta}_{\text{start}}$.

Furthermore, although the term $\bar{L}_{\text{robust}}(\mathbf{x}, y; \theta)$ could in theory be excluded from the training process, as the term $L_{\text{natural}}(\mathbf{x}, y; \theta)$ prevents the degenerate solution of always classifying all images in the abstain class, it's inclusion empirically helps the stability of the training process.

## 5.2 EMPIRICAL RESULTS ON MNIST AND CIFAR10

The classification networks are identical to the large network in Gowal et al. (2018), also detailed in Table 2, trained by minimizing the loss in Eq. 18 with the above stabilizing schemes. Selection of parameters for each datasets is detailed in Appendix B. Since most recent detector networks have shown very low performance against adaptive attacks, and lack provable performance Tramer et al. (2020), we only compare the performance with other provable robust classification methods, while focusing on the different decomposition in the reported natural and robust accuracy among these two. As numbers in Table 1 suggest, the proposed detection/classification network shows improved robustness against other methods, including IBP in isolation (without the detection capability), specially against larger perturbations in the CIFAR-10 dataset, which intuitively is pleasing: as larger perturbations are naturally more distinguishable, the detection capability of the network is successfully leveraged for improving the adversarial robustness. Let us now take a closer look at the performance by focusing on the detection capability.

**Effectiveness of the detection class.** By nature, the proposed classification "adaptively chooses" between (robust) correct classification and detection of *adversarial* or *difficult* inputs during the training. This gives rise to two phenomena:

(1) In verifiably robust methods, natural image accuracy declines as robustness improves. In the proposed approach however, a considerable number of misclassified natural inputs are in fact *abstained on*, which in certain applications is more favorable than assigning them to a wrong class, as classifiers without detection capability would: compare $30.5\%$ abstain and $25.6\%$ 'wrong-class misclassification' (other than abstain and the correct class) in IBP-with-detection, with that of $53.7\%$ 'wrong-class' misclassification in IBP on natural CIFAR-10 images in networks trained for $\epsilon = 8/255$.

(2) Regardless of the training procedure, the proposed classifier with detection can still be verified using verification in Eq. 4 to obtain its guaranteed robustness with only considering the correct class. Thus, comparing this verification percentage with that of Eq. 11 highlights the effectiveness of the abstain class in detecting perturbed images and increasing robustness: for instance, using our method $76.07\%$ maximum robust error successfully decreases to $63.63\%$ by considering the detection capability, on CIFAR-10 trained for $\epsilon = 8/255$, compared to $69.92\%$ in IBP without detection.

| dataset | attack | method | standard err | verified err | pgd-attack-success |
|---|---|---|---|---|---|
| MNIST | $\epsilon_{\text{test}} = 0.3$ $\epsilon_{\text{train}} = 0.4$ | IBP | 2.12 | 8.47 | 6.78 |
| | | IBP w/ detection | 4.34 | **5.98** | **4.15** |
| | | **Best recorded in literature** | | | |
| | | IBP (Gowal et al., 2018) | 1.66 | 8.21 | 6.12 |
| | | IBP-CROWN (Zhang et al., 2020) | 1.82 | 7.02 | 6.05 |
| | | Xiao et al. (2018) | 2.67 | 19.32 | 7.95 |
| | | Mirman et al. (2019) | 2.8 | 11.2 | 4.6 |
| | | Balunovic & Vechev (2020) | 2.7 | 14.3 | – |
| | | Wong & Kolter (2018) | 14.87 | 43.10 | – |
| | $\epsilon_{\text{test}} = 0.4$ $\epsilon_{\text{train}} = 0.4$ | IBP | 2.74 | 14.80 | 11.14 |
| | | IBP w/ detection | 4.79 | **11.29** | **7.55** |
| | | **Best recorded in literature** | | | |
| | | IBP (Gowal et al., 2018) | 1.66 | 15.01 | 10.34 |
| | | IBP-CROWN (Zhang et al., 2020) | 2.17 | 12.06 | 9.47 |
| CIFAR-10 | $\epsilon_{\text{test}} = 2/255$ $\epsilon_{\text{train}} = 2.2/255$ | IBP | 38.54 | 55.21 | 49.72 |
| | | IBP w/ detection | 34.66 | 57.9 | 47.2 |
| | | **Best recorded in literature** | | | |
| | | IBP (Gowal et al., 2018) | 29.84 | 55.88 | 49.98 |
| | | IBP-CROWN (Zhang et al., 2020) | 28.48 | 46.03 | **40.28** |
| | | Balunovic & Vechev (2020) | 21.6 | **39.5** | – |
| | | Mirman et al. (2018) | 38.0 | 47.8 | – |
| | | Wong & Kolter (2018) | 31.72 | 46.11 | – |
| | | Xiao et al. (2018) | 38.88 | 54.07 | 50.08 |
| | $\epsilon_{\text{test}} = 8/255$ $\epsilon_{\text{train}} = 8.8/255$ | IBP | 53.69 | 69.92 | 65.17 |
| | | IBP w/ detection | 55.60 | **63.63** | **49.22** |
| | | **Best recorded in literature** | | | |
| | | IBP (Gowal et al., 2018) | 50.51 | 68.44[+] | 65.23 |
| | | IBP-CROWN (Zhang et al., 2020) | 54.02 | 66.94 | 65.42 |
| | | Mirman et al. (2019) | 43.8 | 72.8 | 65.3 |
| | | Balunovic & Vechev (2020) | 48.3 | 72.5 | – |
| | | Xiao et al. (2018) | 59.55 | 79.73 | 73.22 |
| | | Wong & Kolter (2018) | 71.33 | 78.22 | – |
| | $\epsilon_{\text{test}} = 16/255$ $\epsilon_{\text{train}} = 16.7/255$ | IBP | 68.97 | 78.12 | 76.66 |
| | | IBP w/ detection | 66.37 | **67.92** | **58.20** |
| | | **Best recorded in literature** | | | |
| | | IBP-CROWN (Zhang et al., 2020) | 66.06 | 76.80 | 75.23 |

Table 1: The verified, standard (clean), and PGD attack errors for models trained on MNIST and CIFAR-10. IBP with detection is to be compared with IBP (without detection capability) to emphasize the successful utilization of the detection capability of the network in increasing its verifiable as well as empirical performance. For a more detailed decomposition of the standard and robust error terms see Fig. 1.

[+] As reported in Zhang et al. (2020), achieving the 68.44% IBP verified error requires extra PGD adversarial training loss, without which the verified error is 72.91% (LP/MIP verified) or 73.52% (IBP verified), thus our result should be compared to 73.52%.

* Best reported numbers for IBP are computed using mixed integer programming (MIP), which are strictly smaller than IBP veified error rates, see table 3 and 4 in Gowal et al. (2018). For fair comparison, we report IBP verified error rates from table 3 therein.

** Best reported results from the literature may use different network architectures, and empirical PGD error rate may have been computed under different settings, e.g., number of steps and restarts.

*** Number in the IBP rows in this table are the best between (Zhang et al., 2020) and our experiments, while results from (Gowal et al., 2018) are reported under best literature record for IBP.

† It is important to note that unlike robust classification, the proposed joint classification/detection does successfully leverage the detection capability to decrease the verified error rate by rejecting some adversarial examples, which makes direct comparison of these values difficult. However since there exists no other verifiable detection scheme, such comparison is made here to show the effect of successful detection; see Figure 1 for a detailed discussion on this.

See Fig. 1 for decomposition of the performance metrics of the proposed network over CIFAR-10 dataset, demonstrating the effectiveness of the abstain class in detecting "difficult" natural images while also increasing the robustness certificate over adversarial inputs.

### 5.3 NATURAL VERSUS ADVERSARIAL ERROR TRADEOFF

Reporting a single set point in the Pareto Frontier as reported in Table 1 gives limited understanding on how different methods trade off natural versus robust error. To address this, a more detailed study on this trade-off in IBP-based robust classification with and without detection is discissed here.

In order to get the best performance for IBP-based robust training without detection (that is $\lambda_1 = 0$), and since it is not known whether varying $\kappa_{\text{end}}$ or $\lambda_2$ will lead to a better performance, we have

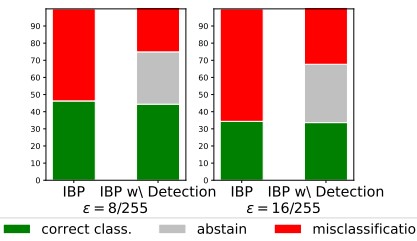

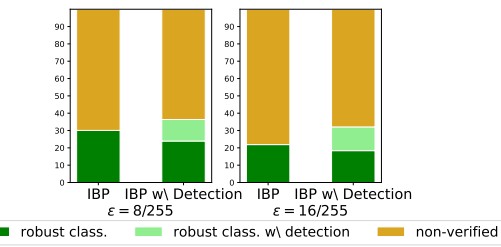

(a) Accuracy on natural (clean) images

(b) Verified accuracy on adversarial images

Figure 1: Decomposition of accuracy and verified accuracy on CIFAR-10 dataste: the detection capability of the network can increase robustness by adaptively abstaining on adversarial inputs while also abstaining on some natural images rather than misclassifying them.

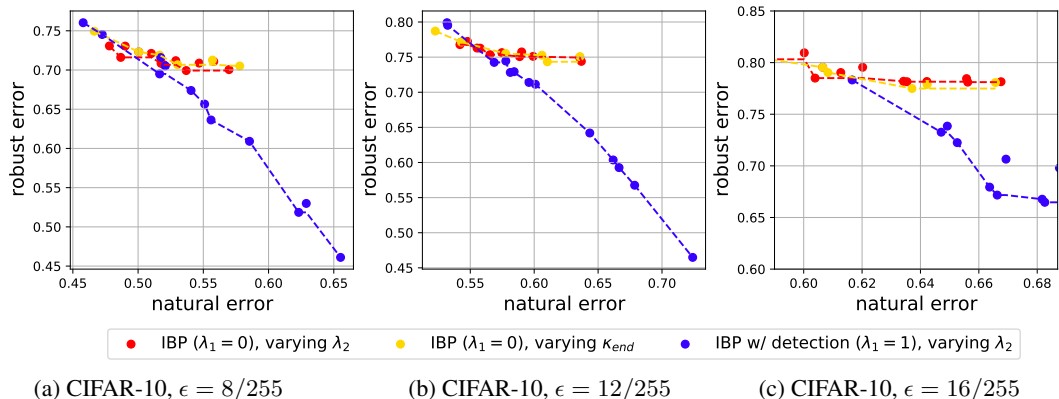

(a) CIFAR-10, $\epsilon = 8/255$          (b) CIFAR-10, $\epsilon = 12/255$          (c) CIFAR-10, $\epsilon = 16/255$

Figure 2: Naural versus robust error tradeoff for IBP ($\lambda_1 = 0$) and IBP-with-detection ($\lambda_1 > 0$) on CIFAR-10 dataset for various perturbation sizes $\epsilon = 8/255, 12/255, 16/255$. Lower curve is better. IBP-with-detection is effectively utilizing its detection capability to adaptively trade natural and robust performance, leading to improved certified robustness against adversarial perturbations.

trained the classification networks in two ways: (1) setting $\kappa_{\text{end}} = 0.5$ and varying $\lambda_2 \in [0, 3]$, and (2) setting $\lambda_2 = 1$ and varying $\kappa_{\text{end}} \in [0, 0.5]$, to get multiple set points along the Pareto Frontier.

Similarly, for IBP-with-detection-based classification, we have set $\kappa_{\text{end}} = 0.5$, $\lambda_1 = 0.6, 0.8, 1.0$ for MNIST and $\lambda_1 = 1.0$ for CIFAR-10, and varied $\lambda_2 \in [1\ 4]$ to get various points along the frontier. The network is trained for various $\epsilon$ values, with other training parameters as stated in Appendix B.

Results are plotted in Fig. 2 and 3 (presented in the Appendix due to space limitation). As shown, the classifier enhanced with detection capability is better able to trade natural and robust accuracy, thus attaining higher robustness by trading small decrease in natural accuracy. This together with the fact that the natural accuracy decrease is also partly handled by abstaining of such natural images that would have been misclassified (as one of the original K classes) otherwise, demonstrates the effective utilization of the detection capability in the proposed method. It is important to note that IBP w/detection allows us to obtain *additional regions* on this Pareto frontier that traditional-robust-classifiers without detection cannot obtain, and could potentially provide additional gain to what is achievable by other various improvement techniques such as tighter relaxation and bound propagation methods.

## 6   CONCLUSION

We proposed a new method for jointly training a provably robust classifier and detector. By introducing an additional "abstain/detection" into a classifier, we have proposed a verification scheme for classifiers with detection under adversarial settings, and shown that such networks can be efficiently trained be extending the common IBP relaxation techniques. The effectiveness of the proposed detection scheme with provable guarantees versus SOTA robust verifiable classification methods is corroborated by empirical tests on MNIST and CIFAR-10, specially against large perturbations.

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

## A   APPENDIX

### A.1   PROOF OF THEOREM 1

Since $\mathcal{Z}_L \in \hat{\mathcal{Z}}_L$ it trivially holds that

$$\min_{\mathbf{z}\in\hat{\mathcal{Z}}_L} \max\{\mathbf{c}_{y,i}^\top\mathbf{z}, \mathbf{c}_{a,i}^\top\mathbf{z}\} \leq \min_{\mathbf{z}\in\mathcal{Z}_L} \max\{\mathbf{c}_{y,i}^\top\mathbf{z}, \mathbf{c}_{a,i}^\top\mathbf{z}\} \tag{20}$$

The lower bound is now a convex minimization, which can be rewritten as

$$\min_{\mathbf{z}\in\hat{\mathcal{Z}}_L} \max\{\mathbf{c}_{y,i}^\top\mathbf{z}, \mathbf{c}_{a,i}^\top\mathbf{z}\} = \min_{\tau,\mathbf{z}\in\hat{\mathcal{Z}}_L} \tau \qquad \text{s. t.} \quad \mathbf{c}_{y,i}^\top\mathbf{z} \leq \tau \quad, \mathbf{c}_{a,i}^\top\mathbf{z} \leq \tau.$$

Defining the slack variables $\eta_a \geq 0$ and $\eta_y \geq 0$ for the inequality constraints, the Lagrangian can be written as

$$\mathcal{L}(\tau, \mathbf{z}, \eta_a, \eta_y) = \tau + \eta_a(\mathbf{c}_{a,i}^\top\mathbf{z} - \tau) + \eta_y(\mathbf{c}_{y,i}^\top\mathbf{z} - \tau)$$

and minimizing $\mathcal{L}(\tau, \mathbf{z}, \eta_a, \eta_y)$ with respect to the primal variable $\tau$, yields $\eta_a + \eta_y = 1$. Defining $\eta := \eta_a = 1 - \eta_y$, and using the fact that the dual maximization always serves as a lower bound on the primal we get

$$\max_{0\leq\eta\leq1} \min_{\mathbf{z}\in\hat{\mathcal{Z}}_L} \left(\eta\,\mathbf{c}_{a,i} + (1-\eta)\,\mathbf{c}_{y,i}\right)^\top \mathbf{z} \leq \min_{\mathbf{z}\in\mathcal{Z}_L} \max\{\mathbf{c}_{y,i}^\top\mathbf{z}, \mathbf{c}_{a,i}^\top\mathbf{z}\}. \square$$

### A.2   PROOF OF THEOREM 2

Following on the statement of Theorem 1 and by substituting $\mathbf{z} = \mathbf{W}_L^\top\mathbf{z}_{L-1} + \mathbf{b}_L$, we get

$$\max_{0\leq\eta\leq1} \min_{\mathbf{z}_{L-1}\in\hat{\mathcal{Z}}_{L-1}} \left(\eta\,\mathbf{c}_{a,i} + (1-\eta)\,\mathbf{c}_{y,i}\right)^\top \left(\mathbf{W}_L^\top\mathbf{z}_{L-1} + \mathbf{b}_L\right) \leq \min_{\mathbf{z}\in\hat{\mathcal{Z}}_L} \max\{\mathbf{c}_{y,i}^\top\mathbf{z}, \mathbf{c}_{a,i}^\top\mathbf{z}\} \tag{21}$$

which can be reordered as

$$\max_{0\leq\eta\leq1} \min_{\underline{\mathbf{z}}_{L-1}\leq\mathbf{z}_{L-1}\leq\bar{\mathbf{z}}_{L-1}} (\boldsymbol{\omega}_1 + \eta\boldsymbol{\omega}_2)^\top\mathbf{z}_{L-1} + \eta\,\omega_3 + \omega_4 \tag{22}$$

where $\boldsymbol{\omega}_1 := \mathbf{W}_L\mathbf{c}_{y,i}$, $\boldsymbol{\omega}_2 := \mathbf{W}_L(\mathbf{c}_{a,i} - \mathbf{c}_{y,i})$, $\omega_3 := \mathbf{b}_L^\top(\mathbf{c}_{a,i} - \mathbf{c}_{y,i})$, and $\omega_4 := \mathbf{b}_L^\top\mathbf{c}_{y,i}$, which then equals

$$\max_{0\leq\eta\leq1} (\boldsymbol{\omega}_1 + \eta\boldsymbol{\omega}_2)^\top\hat{\mathbf{z}}_{L-1} + \eta\,\omega_3 + \omega_4 \tag{23}$$

where minimization w.r.t. $\mathbf{z}_{L-1}$ is solved by the (this is under the setting for most networks with positive activations, and thus lower bound $\underline{\mathbf{z}}_l$ is always non-negative)

$$[\hat{\mathbf{z}}_{L-1}]_j = \begin{cases} [\bar{\mathbf{z}}_{L-1}]_j & \text{if} & [\boldsymbol{\omega}_1 + \eta\boldsymbol{\omega}_2]_j \leq 0 \\ [\underline{\mathbf{z}}_{L-1}]_j & \text{if} & [\boldsymbol{\omega}_1 + \eta\boldsymbol{\omega}_2]_j \geq 0 \end{cases} \tag{24}$$

and can be rewritten as

$$\max_{0\leq\eta\leq1} \sum_{j=1}^{n_{L-1}} \left[\boldsymbol{\omega}_1 + \eta\boldsymbol{\omega}_2\right]_j \left(1_{\{\omega_{1,j}+\eta\omega_{2,j}\leq0\}}[\bar{\mathbf{z}}_{L-1}]_j + 1_{\{\omega_{1,j}+\eta\omega_{2,j}\geq0\}}[\underline{\mathbf{z}}_{L-1}]_j\right) + \eta\,\omega_3 + \omega_4 \tag{25}$$

and can be rewritten as

$$\max_{0\leq\eta\leq1} \sum_{j=1}^{n_{L-1}} \left(1_{\{\omega_{1,j}+\eta\omega_{2,j}\leq0\}}[\boldsymbol{\omega}_1\circ\bar{\mathbf{z}}_{L-1} + \eta\boldsymbol{\omega}_2\circ\bar{\mathbf{z}}_{L-1}]_j + 1_{\{\omega_{1,j}+\eta\omega_{2,j}\geq0\}}[\boldsymbol{\omega}_1\circ\underline{\mathbf{z}}_{L-1} + \boldsymbol{\omega}_2\circ\underline{\mathbf{z}}_{L-1}]_j\right)$$
$$+ \eta\,\omega_3 + \omega_4 \tag{26}$$

where "$\circ$" denotes the elementwise multiplication. Thus, due to the concavity of the dual, optimal $\eta$ can be found by evaluationg the objective in between the break points which are given by $\boldsymbol{\zeta} := [\zeta_1, ..., \zeta_{n_{L-1}}]$ with its j-th element defined as $\zeta_j := -\omega_{1,j}/\omega_{2,j}$.

To do this, let us use $\mathbf{s}$ to denote the $n_L$-ary tuple of indices that sorts $\boldsymbol{\zeta}$. That is

$$\tilde{\boldsymbol{\zeta}} = [\tilde{\zeta}_1, ..., \tilde{\zeta}_{n_{L-1}}] := \Pi_{\mathbf{s}}(\boldsymbol{\zeta}) := [\zeta_{s_1}, ..., \zeta_{s_{n_{L-1}}}] \quad \text{s.t.} \quad \zeta_{s_1} \leq ... \leq \zeta_{s_{n_{L-1}}}$$

with operator $\Pi_{\mathbf{s}}(.)$ denoting the permutation of its arguments according to $\mathbf{s}$, such that $\tilde{\zeta}_i = \zeta_{s_i} \forall i$, and $\tilde{\boldsymbol{\zeta}}$ is sorted in the increasing order .

We can also rewrite the problem by summing over the indices in the sorting set $\mathbf{s}$ instead, as

$$\max_{0 \leq \eta \leq 1} \sum_{j=1}^{n_{L-1}} \left( 1_{\{\omega_{1,j} + \eta \omega_{2,j} \leq 0\}} [\boldsymbol{\omega}_1 \circ \bar{\mathbf{z}}_{L-1} + \eta \boldsymbol{\omega}_2 \circ \bar{\mathbf{z}}_{L-1}]_{s_j} + 1_{\{\omega_{1,j} + \eta \omega_{2,j} \geq 0\}} [\boldsymbol{\omega}_1 \circ \underline{\mathbf{z}}_{L-1} + \boldsymbol{\omega}_2 \circ \underline{\mathbf{z}}_{L-1}]_{s_j} \right)$$
$$+ \eta \, \omega_3 + \omega_4. \tag{27}$$

Now let us define $\underline{\mathbf{u}}_1 := \Pi_{\mathbf{s}}(\boldsymbol{\omega}_1 \circ \underline{\mathbf{z}}_{L-1})$ , $\bar{\mathbf{u}}_1 := \Pi_{\mathbf{s}}(\boldsymbol{\omega}_1 \circ \bar{\mathbf{z}}_{L-1})$, $\underline{\mathbf{u}}_2 := \Pi_{\mathbf{s}}(\boldsymbol{\omega}_2 \circ \underline{\mathbf{z}}_{L-1})$ , $\bar{\mathbf{u}}_2 := \Pi_{\mathbf{s}}(\boldsymbol{\omega}_2 \circ \bar{\mathbf{z}}_{L-1})$, we get

$$\max_{0 \leq \eta \leq 1} \sum_{j=1}^{n_{L-1}} \left( 1_{\left\{ \{\eta \leq \tilde{\zeta}_j \text{ and } \omega_{2,s_j} > 0\} \text{ or } \{\eta \geq \tilde{\zeta}_j \text{ and } \omega_{2,s_j} < 0\} \right\}} (\bar{u}_{1,j} + \eta \bar{u}_{2,j}) \right.$$
$$\left. + 1_{\left\{ \{\eta \geq \tilde{\zeta}_j \text{ and } \omega_{2,s_j} > 0\} \text{ or } \{\eta \leq \tilde{\zeta}_j \text{ and } \omega_{2,s_j} < 0\} \right\}} (\underline{u}_{1,j} + \eta \underline{u}_{2,j}) \right)$$
$$+ \eta \, \omega_3 + \omega_4. \tag{28}$$

In order to break the objective of maximization into piece-wise linear programming subproblems, let us first identify the (indices of) $\zeta_{s_i}$ values that fall in the feasible set $0 \leq \eta \leq 1$ by

$$m = \min_{\zeta_{s_\nu} \geq 0} \nu \quad \text{and} \quad M = \max_{\zeta_{s_\nu} \leq 1} \nu$$

The overall maximization can now be reduced to piece-wise subproblems over sets $\tilde{\zeta}_\nu \leq \eta \leq \tilde{\zeta}_{\nu+1}$ for $m - 1 \leq \nu \leq M$ as

$$\max_{\max\{0, \tilde{\zeta}_\nu\} \leq \eta \leq \min\{1, \tilde{\zeta}_{\nu+1}\}} \sum_{j=1}^{n_{L-1}} \left( 1_{\left\{ \{\eta \leq \tilde{\zeta}_j \text{ and } \omega_{2,s_j} > 0\} \text{ or } \{\eta \geq \tilde{\zeta}_j \text{ and } \omega_{2,s_j} < 0\} \right\}} (\bar{u}_{1,j} + \eta \bar{u}_{2,j}) \right.$$
$$\left. + 1_{\left\{ \{\eta \geq \tilde{\zeta}_j \text{ and } \omega_{2,s_j} > 0\} \text{ or } \{\eta \leq \tilde{\zeta}_j \text{ and } \omega_{2,s_j} < 0\} \right\}} (\underline{u}_{1,j} + \eta \underline{u}_{2,j}) \right)$$
$$+ \eta \, \omega_3 + \omega_4. \tag{29}$$

Since each of these subproblems are maximized at the boundaries of the feasible sets, the overall maximization essentially reduces to evaluation of the following objective function at $(M - m + 3)$ points $\eta = 0, \tilde{\zeta}_m, \tilde{\zeta}_{m+1}, \cdots, \tilde{\zeta}_{M-1}, \tilde{\zeta}_M, 1$

$$g(\eta) = \sum_{j=1}^{n_{L-1}} \left( 1_{\{\omega_{1,j} + \eta \omega_{2,j} \leq 0\}} (\bar{u}_{1,j} + \eta \bar{u}_{2,j}) + 1_{\{\omega_{1,j} + \eta \omega_{2,j} \geq 0\}} (\underline{u}_{1,j} + \eta \underline{u}_{2,j}) \right) + \eta \, \omega_3 + \omega_4$$

Values of $g(\eta)$ can be efficiently computed by a forward cumulative sum and forward-backward cumulative sum of $\underline{\mathbf{u}}_1$ and $\underline{\mathbf{u}}_2$, $\bar{\mathbf{u}}_1$ and $\bar{\mathbf{u}}_2$, thus imposing the overall complexity which is dominated by the sorting at $\mathcal{O}(n_{L-1} \log(n_{L-1}))$ in an efficient implementation. $.\square$

### A.3 DESCRIPTION OF ALGORITHM 1

Here is a step-by-step walk-through for Algorithm 1, with insight on how these steps are performed.

1. Form vectors $\boldsymbol{\omega}_1$ and $\boldsymbol{\omega}_2$, which are the last layer values as $\boldsymbol{\omega}_1 = \mathbf{W}_L \mathbf{c}_{y,i}$, $\boldsymbol{\omega}_2 = \mathbf{W}_L (\mathbf{c}_{a,i} - \mathbf{c}_{y,i})$, $\omega_3 := \mathbf{b}_L^\top (\mathbf{c}_{a,i} - \mathbf{c}_{y,i})$, and $\omega_4 := \mathbf{b}_L^\top \mathbf{c}_{y,i}$.

2. Define $\boldsymbol{\zeta} = [\zeta_1, ..., \zeta_{n_L}] := -\boldsymbol{\omega}_1/\boldsymbol{\omega}_2$ and get the vector of indices $\mathbf{s}$ that sorts it, i.e.,
   $\zeta_{s_1} \leq \cdots \leq \zeta_{s_{n_{L-1}}}$

3. Form the element-wise product of $(\boldsymbol{\omega}_1, \boldsymbol{\omega}_2)$ with $(\underline{\mathbf{z}}_{L-1}, \bar{\mathbf{z}}_{L-1}))$, and sort them according to the index set $s$.
   $\underline{\mathbf{u}}_1 = \Pi_{\mathbf{s}}(\boldsymbol{\omega}_1 \circ \underline{\mathbf{z}}_{L-1}), \bar{\mathbf{u}}_1 = \Pi_{\mathbf{s}}(\boldsymbol{\omega}_1 \circ \bar{\mathbf{z}}_{L-1}), \underline{\mathbf{u}}_2 := \Pi_{\mathbf{s}}(\boldsymbol{\omega}_2 \circ \underline{\mathbf{z}}_{L-1}), \bar{\mathbf{u}}_2 := \Pi_{\mathbf{s}}(\boldsymbol{\omega}_2 \circ \bar{\mathbf{z}}_{L-1}).$

4. Get the lowest and highest indexes $(m, M)$ such that the sorted $\zeta$ vector value at those indices are in the feasible set, between 0 and 1.

5. Iterate over the feasible values of $\eta = 0, \zeta_{s_m}, \zeta_{s_{m+1}}, \cdots, \zeta_{s_{M-1}}, \zeta_{s_M}, 1$ and compute the corresponding objective values

$$g(\eta) = \sum_{j=1}^{n_{L-1}} \left( 1_{\{\omega_{1,j} + \eta\omega_{2,j} \leq 0\}} \left( \bar{u}_{1,j} + \eta\bar{u}_{2,j} \right) + 1_{\{\omega_{1,j} + \eta\omega_{2,j} \geq 0\}} \left( \underline{u}_{1,j} + \eta\underline{u}_{2,j} \right) \right)$$
$$+ \eta\,\omega_3 + \omega_4$$

6. Return the maximum value of $g(\eta)$ over the evaluated points.

## A.4 PROOF OF THEOREM 3

Let us start by splitting the feasible set into disjoint sets of

$$\hat{\mathcal{Z}}_{L-1}^a := \{\mathbf{z}_{L-1} \mid z_{L-1,a} \geq z_{L-1,y}\}, \text{ and } \hat{\mathcal{Z}}_{L-1}^y := \{\mathbf{z}_{L-1} \mid z_{L-1,a} < z_{L-1,y}\}$$

where

$$\hat{\mathcal{Z}}_{L-1} = \hat{\mathcal{Z}}_{L-1}^y \cup \hat{\mathcal{Z}}_{L-1}^a, \text{ and } \hat{\mathcal{Z}}_{L-1}^y \cap \hat{\mathcal{Z}}_{L-1}^a = \emptyset.$$

Proof is carried out by considering $\mathbf{z} \in \hat{\mathcal{Z}}_{L-1}^y$ and $\mathbf{z} \in \hat{\mathcal{Z}}_{L-1}^a$, separately.

Restricting $\mathbf{z} \in \hat{\mathcal{Z}}_{L-1}^y$ we have $\ell_{\text{xent}\backslash a}(f_\theta(\mathbf{x} + \boldsymbol{\delta}), y) \leq \ell_{\text{xent}\backslash y}(f_\theta(\mathbf{x} + \boldsymbol{\delta}), a)$ which leads to

$$L_{\text{robust}}^{\text{abstain}}(\mathbf{x}, y; \theta) = \max_{\boldsymbol{\delta} \in \Delta} \min \left\{ \ell_{\text{xent}\backslash a}(f_\theta(\mathbf{x} + \boldsymbol{\delta}), y), \ell_{\text{xent}\backslash y}(f_\theta(\mathbf{x} + \boldsymbol{\delta}), a) \right\} \tag{30}$$

$$\leq \max_{\mathbf{z}_{L-1} \in \hat{\mathcal{Z}}_{L-1}^y} \ell_{\text{xent}\backslash a}(\mathbf{z}_L, y) \quad \text{s.t.} \quad \mathbf{z}_L = \mathbf{W}_L^\top \mathbf{z}_{L-1} + \mathbf{b}_L \tag{31}$$

Loss function $\ell_{\text{xent}\backslash a}$ is the cross entropy loss defined on the $K$-dimensional vector $[z_{L,1}, \cdots, z_{L,K}]$ and class $y$, and thus following Wong & Kolter (2018) given its transnational invariance equals

$$\max_{\mathbf{z}_{L-1} \in \hat{\mathcal{Z}}_{L-1}^y} \ell_{\text{xent}\backslash a}(\mathbf{z}_L, y) = \max_{\mathbf{z}_{L-1} \in \hat{\mathcal{Z}}_{L-1}^y} \ell_{\text{xent}\backslash a}(\mathbf{z}_L - z_{L,y}\mathbf{1}, y) \quad \text{s.t.} \quad \mathbf{z}_L = \mathbf{W}_L^\top \mathbf{z}_{L-1} + \mathbf{b}_L \tag{32}$$

with $\mathbf{1}$ denoting the $(K+1)$-dimensional vector of all ones. Given the invariance of $\ell_{\text{xent}\backslash a}$ with respect to $z_{L,a}$, it can finally be upperbounded by taking the upperbound for all $i$ indices where $i = 1, ..., K, i \neq a, y$ and lowerbound at index $i = y$. Note that for $i = y$, value $[\mathbf{z}_L - z_{L,y}\mathbf{1}]_i = 0$, and a lower bound on other entries $i = 1, ..., K, i \neq a, y$ can be obtained by

$$z_{L,i} - z_{L,y} = -\max\{z_{L,y} - z_{L,i}, z_{L,a} - z_{L,i}\} = -\max\{\mathbf{c}_{y,i}^\top \mathbf{z}, \mathbf{c}_{a,i}^\top \mathbf{z}\} \tag{33}$$

$$\leq -\min_{\mathbf{z}_L \in \mathcal{Z}_L} \max\{\mathbf{c}_{y,i}^\top \mathbf{z}, \mathbf{c}_{a,i}^\top \mathbf{z}\} \leq -J_i(\mathbf{x}, y) \leq -J_i^{\eta, \bar{\eta}}(\mathbf{x}, y) \tag{34}$$

where the first equality holds since $\hat{\mathcal{Z}}_{L-1}^y := \{\mathbf{z}_{L-1} \mid z_{L-1,a} < z_{L-1,y}\}$ for $\mathbf{z} \in \hat{\mathcal{Z}}_{L-1}^y$, second inequality is due to Theorem 2, and third inequality is given by Eq. 15.

Thus, for $\mathbf{z} \in \hat{\mathcal{Z}}_{L-1}^y$ the loss term is now upperbounded by

$$L_{\text{robust}}^{\text{abstain}}(\mathbf{x}, y; \theta) \leq \ell_{xent\backslash a}(-\mathbf{J}_{\epsilon,\theta}(\mathbf{x}, y), y)$$

where

$$[\mathbf{J}_{\epsilon,\theta}(\mathbf{x}, y)]_i = \begin{cases} 0 & \text{if} \quad i = a, y \\ J_i^{\eta, \bar{\eta}}(\mathbf{x}, y) & \text{otherwise.} \end{cases} \tag{35}$$

| Network layers | |
|---|---|
| Conv 64 $3 \times 3 + 1$ | |
| Conv 64 $3 \times 3 + 1$ | |
| Conv 128 $3 \times 3 + 2$ | |
| Conv 128 $3 \times 3 + 1$ | |
| Conv 128 $3 \times 3 + 1$ | |
| Fully Conn. | 512 |
| # hidden | 230K |
| # params. | 17M |

Table 2: Network architecture. Similar to the Large network used in (Gowal et al., 2018)

Similarly, it can be shown that for Thus, for $\mathbf{z} \in \hat{\mathcal{Z}}^a_{L-1}$ the loss term is now upperbounded by

$$L^{\text{abstain}}_{\text{robust}}(\mathbf{x}, y; \theta) \leq \ell_{\text{xent}\backslash y}(-\mathbf{J}_{\epsilon,\theta}(\mathbf{x}, y), a).$$

The equality of $\ell_{\text{xent}\backslash y}(-\mathbf{J}_{\epsilon,\theta}(\mathbf{x}, y), a) = \ell_{\text{xent}\backslash a}(-\mathbf{J}_{\epsilon,\theta}(\mathbf{x}, y), y)$ trivially follows from the fact that $[\mathbf{J}_{\epsilon,\theta}(\mathbf{x}, y)]_i = 0$ for $i = a, y$.

Thus, since $\hat{\mathcal{Z}}_{L-1} = \hat{\mathcal{Z}}^y_{L-1} \cup \hat{\mathcal{Z}}^a_{L-1}$, the proof is complete. $\qquad\Box$

## B  APPENDIX: EXPERIMENT SET UP

Training parameters and schedules are similar to (Gowal et al., 2018) and (Zhang et al., 2020), and outlined in detail here. For training the classifier network with architecture given in Table 2, for both datasets, Adam optimizer with learning rate of $5 \times 10^{-4}$ is used. Unless stated differently, $\kappa$ is scheduled by a linear ramp-down process, starting at 1, which after a warm-up perio,d is ramped down to value $\kappa_{\text{end}} = 0.5$. Value of $\epsilon$ during the training is also simultaneously scheduled by a linear ramp-up, starting at 0, and ramped up to the final value of $\epsilon_{\text{train}}$, reported in Tabel 1, and networks are trained with a single NVIDIA Tesla V100S GPU.

- For MNIST, the network is trained in 100 epochs with batchsize of 100 (total of $60K$ steps). A warm up period of 3 epochs ($2K$ steps) is used (normal classification training with no robust loss), followed up by a ramp-up duration of 18 epochs ($10K$ steps), and the learning rate is decayed $\times 10$ at epochs 25 and 42. No data augmentation is used. Furthermore, fixed selection of $\bar{\eta} = 0.9$ and $\underline{\eta} = 0.1$ during training is used for this dataset with no ramp-down. Reported numbers in Table 1 corresponds to $\lambda_1 = 1$ and $\lambda_2 = 2$ for $\epsilon = 0.3$, and $\lambda_1 = 0.6$ and $\lambda_2 = 1$ for $\epsilon = 0.4$ respectively.

- For CIFAR10, the network is trained in 3200 epochs with batchsize of 1600 (total of $100K$ steps). A warm up period of 320 epochs ($10K$ steps) is used (normal classification training with no robust loss), followed up by a ramp-up duration of 1600 epochs ($50K$ steps), and the learning rate is decayed $\times 10$ at epochs 2600 and 3040 (60k and 90K steps). Random translations and flips, and normalization of each image channel (using the channel statistics from the train set) is used during training. Furthermore, during training for all $\epsilon$ values we have selected $\bar{\eta}_{\text{start}} = 1.0$ and $\bar{\eta}_{\text{end}} = 0.9$. Additionally, $\underline{\eta}_{\text{end}} = 0.1$ is used during training, with $\underline{\eta}_{\text{start}} = 0.1$ for $\epsilon = 2/255$ (no ramp down), $\underline{\eta}_{\text{start}} = 0.3$ for $\epsilon = 8/255$, $\underline{\eta}_{\text{start}} = 0.4$ for $\epsilon = 12/255$, and $\underline{\eta}_{\text{start}} = 0.5$ for $\epsilon = 16/255$. The intuition behind these parameters selection lies in Remark 2, as large $\eta$ values promote the abstain option more, so for large $\epsilon$, we start with larger $\underline{\eta}_{\text{start}}$ as well. Reported numbers in Tabel 1 correspond to $\lambda_1 = 1$ for all $\epsilon$ values, and $\lambda_2 = 3.0$ for $\epsilon = 2/255$, $\lambda_2 = 2.9$ for $\epsilon = 8/255$, and $\lambda_2 = 3.1$ for $\epsilon = 16/255$ to insure similar natural accuracy for fair comparison against other methods.

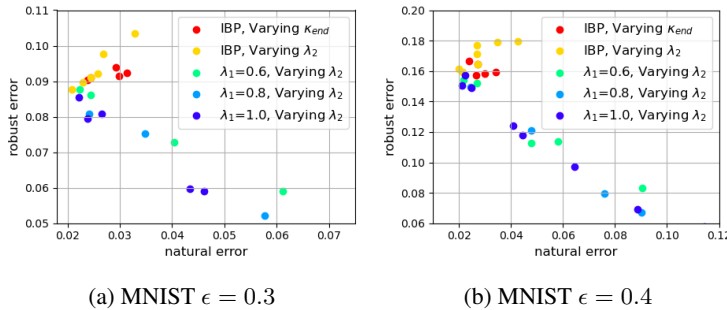

(a) MNIST $\epsilon = 0.3$         (b) MNIST $\epsilon = 0.4$

Figure 3: Naural versus robust error tradeoff for IBP ($\lambda_1 = 0$) and IBP-with-detection ($\lambda_1 > 0$) on MNIST dataset for various perturbation sizes $\epsilon = 0.3$ and $\epsilon = 0.4$. Points closer to the origin are better. IBP-with-detection is effectively utilizing its detection capability to adaptively trade natural and robust performance, leading to improved certified robustness against adversarial perturbations.

## B.1   PARETO FRONTIER FOR MNIST DATASET

## B.2   EMPIRICAL ATTACK SUCCESS RATE USING PGD ATTACKS

In order to obtain empirical attack success on the trained networks, adversarial perturbations are sought by solving

$$\max_{\boldsymbol{\delta} \in \Delta_\epsilon} \left( \max_{i \neq a, y} z_{L,i} - \max\{z_{L,y}, z_{L,a}\} \right) \tag{36}$$

This attack is indeed an adaptive attack as it aims at circumventing the detection while trying to cause misclassification (Tramer et al., 2020). Perturbations are sought by maximizing this objective using PGD with 200-steps for mnist and 500-steps for CIFAR-10 Madry et al. (2017), with 10 random restarts. It is interesting to note that the achieved attack success rate in Table 1 is well below the verified robust error, further implying the effectiveness of incorporation of the detection mechanism as the true robustness of the system against practical adaptive PGD attacks are considerably stronger in comparison to robust classification without detection.

