# OpenReview forum: "Provably robust classification of adversarial examples with detection"
_ICLR.cc/2021/Conference — ICLR 2021 Poster_

### Official Review · AnonReviewer2 · 2020-10-24
**Missing comparison to related work and experiments.**

**Rating:** 5
**Confidence:** 3

**Review:**

This paper aims to train networks that can map a possibly $\ell_\infty$-perturbed input to its class provably or map this input to the “abstain class” provably. This is achieved by training on the IBP output boxes together with a new loss function. The method diverts from the classical setting in which the classifier needs to be robust, by allowing the classifier to abstain. While this is an interesting idea, the paper seems to be rushed and not carefully written. The notation could be improved and presentation could be simplified. It seems as if the results from related work (Balunovic et. al. [3] and Mirman et. al. [4]) are missing in Table 1 and thus are not compared against. Further, some questions remain open after reading the paper:

Questions:
- Would the idea to have an explicit “abstain” together with the loss function work also for randomized smoothing [1]? How would the results look like?
- Have you tried to use certification methods using tighter relaxations like k-ReLU[2]?
- In Equation 8: Is this here for all $i$?
- How would your method perform when using COLT [3]?

Comments:
- in equation 3, it seems to me as if the second ‘+’ for $\underline{z}_l$ should be a ‘-’, similar for $\overline{z}_l$
- in equation 6: $\ell_{\text{xent}}$ should be defined before used

The comparison in the evaluation is not complete, questions remain open after reading the paper and evaluation is missing, hence this is a reject for me.


[1] Cohen et. al.: 'Certified Adversarial Robustness via Randomized Smoothing'. http://proceedings.mlr.press/v97/cohen19c/cohen19c.pdf

[2] Singh et. al.: 'Beyond the Single Neuron Convex Barrier for Neural Network Certification'. http://papers.nips.cc/paper/9646-beyond-the-single-neuron-convex-barrier-for-neural-network-certification.pdf

[3] Balunovic et. al.: 'Adversarial Training and Provable Defenses: Bridging the Gap'. https://openreview.net/pdf?id=SJxSDxrKDr

[4] Mirman et. al.: 'A Provable Defense for Deep Residual Networks'. https://arxiv.org/pdf/1903.12519.pdf

######################

After reading the author's response: While i still think that extending the experimental evaluation along the axes described above would improve this work, i decided to raise my score.

---

> ### Author Response · Authors · 2020-11-14
> **Thank you for your thoughtful questions and comments.**
>
> We thank the reviewer for their comments and feedback. The revised manuscript is now updated to include the performance of the suggested prior work for a more thorough comparison, and the comments and questions are addressed next in detail.
>
> Q: “1. Would the idea to have an explicit “abstain” together with the loss function work also for randomized smoothing [1]? How would the results look like?”
>
> A: This is an interesting question. Randomized smoothing is among the successful robustification approaches available in the literature, where adding randomness to the input helps in exploring the neighborhood of the image, and the corresponding “smooth classifier” can be used to provide robustness guarantees against adversarial perturbations. This method however, differs considerably with the line of research that has been the focus of this paper, where bound propagation is used for robustness certification. Due to the usage of such considerably different techniques, it is not clear how one can incorporate an abstain option with randomized smoothing. We appreciate the reviewer for this comment, and we think this is a very interesting direction to investigate.
>
>
> Q: “2. Have you tried to use certification methods using tighter relaxations like k-ReLU[2]?”
>
> A: As correctly pointed out, an active line of research is in fact targeted at providing tighter relaxation schemes for improved verification. Namely, K-ReLU[2] and similar approaches therein, while providing tighter relaxation by jointly considering multiple activations in a hidden-layer, impose a considerable complexity to the propagation process, thus rendering such methods difficult to incorporate during the training phase. Applying such methods on the testing process on the other hand, although could lead to tighter verified accuracy, should similarly be done for all of the discussed robust classification methods to provide a fair comparison, making such comparison however is very difficult, as different methods could also have different optimal tighter-relaxation counterparts. In addition, the contribution of this work is focused on investigating the benefits of an effectively trained detection class, rather than providing tighter relaxation methods. Thus, we have refrained from reporting verified accuracy obtained by the more expensive tighter verification methods, such as Mixed-integer Programming (MIP) or K-Relu [2] for any of the reported methods to have a fair comparison, and aimed at disentangling the effects of tighter relaxation from that of the detection capability itself.
>
>
> Q: “3. In Equation 8: Is this here for all i ?”
>
> A: Yes. We have updated the manuscript to clarify this.
>
>
> Q: “4. How would your method perform when using COLT [3]?”
>
> A: The interesting method proposed in COLT [3] aims at bridging the gap between adversarial training and verification-based training techniques by utilizing the propagated interval bounds in order to craft adversarial examples in the latest space (hidden layers) of the network. Numerical evaluations have shown COLT to be effective in providing improvements in terms of the verified accuracy; however such improvements seem to be limited for adversarial settings with small $\epsilon$, e.g. $\epsilon = 0.1$ for MNIST and $\epsilon=2/255$ for CIFAR-10, while the provided accuracy is worse for higher values of epsilon, namely for $\epsilon = 0.3$ for MNIST and $\epsilon=8/255$ for CIFAR-10; see [3]. This is however in contrast to the proposed joint/classification method, where empirical results indicate the detection capability to be advantageous for higher $\epsilon$ values, that is $\epsilon >= 8/255$ for CIFAR for instance. This is also intuitively pleasing as adversarially perturbed images with higher values of epsilon have a bigger difference with their natural counterparts than those with smaller values of epsilon, and thus are possibly easier to be detected. Hence, given the effectiveness of detection at higher epsilon values, and lower performance of COLT in those regimes, [3] does not fit as a promising method to be incorporated for these regimes, and the verification-based training process is a better fit.
>
> Comments:
>
>
> Q: “In equation 3, it seems to me as if the second ‘+’ for z_l  should be a ‘-’, similar for z¯l in equation 6”
>
> A:Thank you for pointing this out. The updated manuscript is corrected.
>
> Q: “ℓxent  should be defined before used”
>
> A: Thank you for this comment. It is now defined before its usage in the updated manuscript.

---

> > ### Author Response · Authors · 2020-11-14
> > **Authors' Response, Part 2**
> >
> > We would like to thank the reviewer for their thoughtful comments, and acknowledge that the above points are interesting as they explore new directions in which the proposed certifiable joint classification/detection could be extended. However, the scope of this work has mainly focused on providing an eye-to-eye comparison with IBP- that is fixing the bound propagation and certification method- and we have aimed at disentangling the effects of more expensive and tighter relaxation/certification techniques, and thus focused on what the detection capability itself can bring about. We hope this can also show the merits and advantages of the proposed work, as its validity motivates further exploration in the above directions.
> > Given the numerous work in the area and variety of available techniques for verifiable classification as well as relaxation/bound-propagation methods, a thorough investigation of these methods, and naturally optimizing such techniques for the proposed joint classification/detection is an interesting research direction, and will be explored in our future work.
> >
> > We hope that the raised concerns have been fully addressed, and we would be grateful to discuss any remaining concerns in more detail. We look forward to receiving your feedback and comments.

---

### Official Review · AnonReviewer4 · 2020-10-27
**Provably robust classification of adversarial examples with detection**

**Rating:** 6
**Confidence:** 5

**Review:**

This paper proposes a new defense method based on adding a reject class to a classifier trained for robust classification. The paper has merits, it is formally sound and it improves the SOA. However, my rating is not so high for the following reasons:

- Only a short paragraph at the end is devoted to the natural-vs-robust error trade-off and the comparison is done only with IBP (another defense method of 2018). In my opinion this important trade-off is being neglected. What is the point of developing classifiers that are robust to AEs when they are not robust to other normal images (like those found in the test set)?
- Adding a reject class may not seem fair. Particularly, the paper states (p.4) "the classification/detection tasks is considered successful if the input is classified either as the correct class y or as the abstain class a; as both cases prevent misclassification of the adversarially perturbed input as a wrong class". I do not agree with that; it should be considered successful only if the input is classified as the correct class y, period. The "abstain" class is just a reject option, which is not the same as the correct classification. One can always resort to rejection to improve the error rate in the non-rejected part of the dataset. When there is a reject option in the system the figure of merit should be the ratio between error and rejection rate, while in your case you are using only error as the figure of merit. What would be the error-reject trade-off curve of the base classifier in these same datasets?

Also, I could not understand the relative weights lambda 1 and 2 in Eq. 14. Are they correct? shouldn't there be a weight also for the first term Lrobust? Likewise in Eq. 18 you state that for lambda1=0 the training reduces to that of Gowal but the weight terms are different in Eqs. 18 and 7.

Minor:
- When referencing an equation you should capitalise the first letter E.
- "plotted in 2"
- "network can increase robust"

---

> ### Author Response · Authors · 2020-11-10
> **Misplaced review**
>
> We thank the reviewer for their time, however, it seems like the comments correspond to another submission, and the ones under Reviewer 4 in the following submission correspond to our work.
>
> https://openreview.net/forum?id=4mkxyuPcFt&noteId=ceP75ed9eB
>
> We'd appreciate it if you look into this. Thanks.

---

> > ### Comment · AnonReviewer4 · 2020-11-10
> > **Yes**
> >
> > oops, yes you're right, I reviewed both papers but misplaced the reviews, I'm so sorry.
> > I'll see what I can do to fix it

---

> > > ### Author Response · Authors · 2020-11-10
> > > **Thanks**
> > >
> > > Great. Thanks for the prompt response. We will be responding to the corresponding comments soon.

---

> > > > ### Comment · AnonReviewer4 · 2020-11-11
> > > > **corrected**
> > > >
> > > > this has been corrected, thanks again for noticing it

---

> ### Author Response · Authors · 2020-11-14
> **Authors' Response Part 1/2**
>
> We thank the reviewer for their thoughtful comments, and recognizing the merits and improvements over SOTA brought about by the proposed method. The raised concerns are addressed next.
>
> Q: “ Only a short paragraph at the end is devoted to the natural-vs-robust error trade-off…”
>
> A: We agree on the fact that natural-vs-robust error trade-off is an important aspect of robust classification, and reporting single numbers out of this curve only partially depicts the performance of a given method. In the updated manuscript, we have now included a more detailed study on the trade-off, and the curves for different values of $\epsilon$ for MNIST and CIFAR by varying trade-off parameters $\lambda_1$ and $\lambda_2$ are plotted in Fig 2. and 3- these figures were previously in the Appendix due to space limitation, but now have been emphasized more in the main body of the work. These curves also further delineate the fact that the proposed abstain/rejection option can indeed utilize its detection capability to better trade natural and robust performance, leading to improved curve compared to a classifiers without the detection capability.
>
> There is also a line of research in which the trade-off between natural accuracy and adversarial robustness is statistically studied [1, 2], however, the trade-off is still far from perfect considering the sacrificed natural image accuracy, keeping this line of research open to new ideas for a better trade-off. We believe provable detection is among such schemes, and further investigations can help advance the area.
>
>
>
> Q: “Adding a reject class may not seem fair….”
>
> A: This is indeed a good point, as we also have mentioned in various occasions in the manuscript, clarifying that the robust classification methods are not directly comparable with the proposed jointly robust classification/detection. This is due to the fact that in the presence of a detector, the definition of a successful attack inevitably changes, as an attack is considered now successful only if the perturbed image evades detection and causes misclassification, and violation of either of these two renders an attack unsuccessful. Indeed, there may be applications for which detection of adversarial inputs bears no value in itself, for instance if requiring human annotation in case of detection incurs a high cost or is infeasible due to restricted latency requirements. In such cases, robust classification seems to be the only viable option for providing robustness. In this work however, we have focused on the case where detection of adversarial inputs does bring value to the underlying task, and we have aimed at studying whether adversarial examples can be successfully detected and if such a detection capability can be rigorously verified.
>
> Thus inherently, results and performance metric of a joint classification/detection differs with that of robust classification. Unfortunately, there is no other detection scheme in the literature with provable guarantees to compare the proposed method against. In fact, previous works on detection such as [3] have also compared with robust classification methods in terms of robust error as it is the only viable benchmark - despise the fact that even [3] was also later shown to be mostly ineffective against carefully crafted adaptive attacks in [4], further emphasizing the need for provable guarantees.
>
> To emphasize this difference in performance, the footnote of Table 1 has been updated to explicitly clarify this point. In addition, Figure 1 also provides a detailed decomposition and comparison of the accuracy/abstain ratio on the natural as well as adversarial images for robust classification and joint classification/detection. We show that (a) the joint classification/detection is also certifiable when only considering the correct class as the acceptable one- although providing a slightly smaller value than that of robust classification- (b) the proposed method also provides improved certification if the definition is extended to accept abstain/rejection for perturbed images. (c) Interestingly, the abstain class also helps with natural images as well by abstaining on some of them, rather than making a wrong classification decision (classifying as a wrong class among the original K classes) as it happens in the robust classification. That is however, depending on application, if an abstain is considered as a more desirable outcome than a misclassification (to one of the original K classes) for natural images.
>
> Furthermore, regarding the trade-off curve, as mentioned “One can always resort to rejection to improve the error rate in the non-rejected part of the dataset.”; however, the challenge is to prevent rejection on clean images as resorting to rejecting on the non-rejected part of the dataset reduces the natural image accuracy. Thus, we have provided natural-vs-robust error curves in the manuscript to address this trade-off.

---

> > ### Author Response · Authors · 2020-11-14
> > **Authors' Response Part 2/2**
> >
> > Q: “Also, I could not understand the relative weights lambda 1 and 2 in Eq. 14...”
> >
> > A: Thanks for your comment. Tuning of parameters $(\lambda_1, \lambda_2)$ plays a critical role in the tradeoff between natural and adversarial accuracies; for instance see [2] for a theoretical study on the trade-off is a similar setting. To clarify such effects let us consider a few corner cases: (a) Setting $\lambda_1 \leftarrow 0$  and a very large $\lambda_2$ the trained classifier tends to a standard (non-robust) classifier, (b) selecting a very large $\lambda_1 \gg 0$ and $\lambda_2 \leftarrow 0$, the model approaches an always-abstaining degenerate classifier, and (c) setting  $\lambda_1 \leftarrow 0$ would render the overall loss similar to that in IBP where natural and robust classification are traded off by parameter $lambda_2$ and the abstain class is never promoted. However, as correctly pointed out, for the loss terms in Eq. (18) and (7) to be exactly equal further modification is needed: the second term in Eq. (7) should also be weighted with parameter $\gamma$, and by setting $\gamma=\lambda_2$, and $\kappa=0.5$, this comparison would hold. This has been corrected in the manuscript; thanks for pointing this out.
> > Furthermore, it is possible to introduce a weight parameter for the $L_\text{robust}$ term as well, however normalizing with respect to any one of the three terms in the loss can effectively capture the same trade-off effect and reduce parameter tuning to two parameters, and thus we have done so by normalizing the weight parameter of the term $L_\text{robust}$; and assigning weight 1 to it.
> >
> > In Subsection 5.3 we empirically study the impact of these weight parameters in detail, where we have trained the network for a range of $\lambda_1$ and $\lambda_2$ values, and the resulting curve is plotted for comparison. It is also interesting to note that the proposed algorithm can better trade off the natural and adversarial errors in comparison with classification-only robust networks, perhaps due to effective utilization of the detection capability.
> >
> >
> > Q: “Minor comments"
> >
> > A:Thanks for pointing these out. The updated manuscript is corrected accordingly.
> >
> > We hope that the raised concerns have been fully addressed, and we would be grateful to discuss any remaining points in more details as needed. We look forward to receiving your feedback and comments.
> >
> >
> > [1] Dimitris Tsipras, Shibani Santurkar, Logan Engstrom, Alexander Turner, and Aleksander Madry. Robustness may be at odds with accuracy. In International Conference on Learning Representations, 2019
> >
> > [2] Zhang, Hongyang, et al. "Theoretically principled trade-off between robustness and accuracy." arXiv preprint arXiv:1901.08573 (2019).
> >
> > [3] Yin, Xuwang, Soheil Kolouri, and Gustavo K. Rohde. "GAT: Generative Adversarial Training for Adversarial Example Detection and Robust Classification." International Conference on Learning Representations. 2020.
> >
> > [4] Tramer, F., Carlini, N., Brendel, W. and Madry, A., 2020. On adaptive attacks to adversarial example defenses. arXiv preprint arXiv:2002.08347.

---

### Official Review · AnonReviewer1 · 2020-10-29
**Good paper, combining verification methods and adversarial example detection**

**Rating:** 7
**Confidence:** 4

**Review:**

Summary:
This paper deals with the problem of bounding the amount of errors that a model can make when attacked by an adversary limited to small perturbation of an input image. Similarly to previous paper, it proposes to extend the classifier with a detector to identify adversarial examples. As opposed to previous papers, the authors suggest to perform formal verification to prove that the samples are either robustly classified or abstained on (as opposed to testing the classifier + detector adversarially which would have left open the question of whether or not the attack against the pair was done in an adequate way). The method used for this is based on the simple IBP algorithm but the authors propose a proper encoding of the max(abstain_logit, gt_logit). They also follow the IBP strategy of using the bound resulting from verification as part of the training objective in order to encourage the network to behave properly.
Experimental results are reported on MNIST and CIFAR-10, and shows that there are scenarios in which the behaviour caused by this algorithm might be desirable.The experimental protocol for training network with the proposed method is described in significant amount of details and it should be possible to reproduce experimental results based on the description.

The crux of the paper lies in redefining the way accuracy should be computed in the case of an abstain option and adjusting verification and robust training methodologies for it. The accuracy is computed in the following way:
* For natural images, the prediction is considered correct only if the output is the correct class.
* For perturbed image, the prediction can be either the correct class OR the abstain class.
This might be sensible, but unless I'm mistaken, this could lead to degenerate cases where the network could achieve a verified error of 0 while having a standard error of 100% (see below). It's regrettable that we lose the property of the verified error rate being an upper bound on the standard error rate.

If I'm not mistaken, the results in Table 1 are hard to evaluate because the methods proposed by the authors is the only one that considers an abstain class, which gives it leeway to "cheat". A pathological example of this would be a network always outputting "abstain". If I understand correctly, it would be classified as having 0% verified error, while having 100% standard error. A moderate version of this is what can be observed for all the pairs (except CIFAR10-eps=16) : standard error goes up compared to IBP while verified error goes down.
Figure 1. is however quite helpful in showing that there might be some benefit in the proposed approach if the goal of an application leans more towards "Never making bad predicitions" rather than "Always making good predicitions".

Comments:
- I am not entirely sure of the premise of the papers that "adversarial examples is something that we can defend against". I am not convinced that people are worried about an attacker crafting attacks to fool their classifier. The problem is that we want our classifiers to be robust to small perturbation, because that's a desirable property and adversarial examples is a sign that this property is not met.

There might be some potential extension to this paper in the direction of OOD detection. Out of Distribution examples tend to make the classifier perform poorly. There is also a glut of methods that claims to be able to detect out of distribution examples without any formal guarantees so I would encourage the authors to look in that direction for extensions to their work.

Opinion: The paper is quite interesting to read, and as far as I can tell is the first one to apply verification methods to the detection of adversarial examples. This is a welcome contribution as opposed to all the methods that claim to detect attacks but can not provide any guarantees.

---

> ### Author Response · Authors · 2020-11-14
> **Thank you for your thoughtful comments and encouraging remarks**
>
> We thank the reviewer for their thorough and thoughtful comments, and for their recommendation for acceptance. We also appreciate the suggestions for possible extensions of the work, and also recognizing the merits of the proposed approach.
> In particular, regarding the comment on the fact that adversarial examples are a sign that the desired robustness property of the available classifiers are not met, we share the same opinion, as robustification against them leads to losing performance on clean images. As shown in Fig 2 and 3 of the updated manuscript, we have demonstrated that the natural-vs-robust accuracy tradeoff seems to have plenty of room for improvement in the available methods, while the proposed approach of incorporating a detection capability can improve the curve. There is also a line of research in which the trade-off between natural accuracy and adversarial robustness is statistically studied [1, 2], however, the trade-off is still far from perfect considering the sacrificed natural image accuracy, and we also believe that further improvements are of high interest to improve this aspect. However, until now, robustification against adversarial examples seems to be among the more well-defined metrics for rigorously measuring robustness, although other definitions of robustness against OOD data are also under investigation in the community, and could have an application for our detection scheme, as correctly suggested, as well.
>
>
> [1] Dimitris Tsipras, Shibani Santurkar, Logan Engstrom, Alexander Turner, and Aleksander Madry. Robustness may be at odds with accuracy. In International Conference on Learning Representations, 2019
>
> [2] Zhang, Hongyang, et al. "Theoretically principled trade-off between robustness and accuracy." arXiv preprint arXiv:1901.08573 (2019).

---

### Official Review · AnonReviewer3 · 2020-11-03
**Borderline**

**Rating:** 5
**Confidence:** 3

**Review:**

##########################################################################

Summary:

In this paper, the authors propose an additional "abstain/detection" loss term into training, so that the classifier can either robustly classify or detect an adversarial attack. They extend the interval bound propagation method for certified robustness under L_infity perturbations (a simple bounding in changes of the weight after each NN layer application, using a box constraint). Hyperparameters in the objective trade-off between clean and adversarial accuracy.

##########################################################################

Reasons for score:

Overall, I think the paper is borderline. I like the idea of adding an "abstain" error term in the loss during training, and I like that the authors have derived some (simple) bounds on the min-max formulations. They show some benefit of addition of abstain/detection in the training. However, my major concern is that if these results could also be obtained without adding these losses in the objective, but by simply comparing "closeness" or confusion in the class labels from the classifier, and abstaining from classification as a post-processing step instead. What is the benefit of adding this to the training?

##########################################################################

Pros:

The authors are trying to break the loop of adversarial training and subsequent adaptive attacks, by developing "provably robust" methods.

The idea of incorporating an "abstain" loss term at the training stage, and trading it off with the true error in classification is interesting. They further modify the loss slightly to exclude true and abstain classes (perhaps, driven by experiments).

They obtain (simple) convex upper bounds on the loss functions and minimize these.

##########################################################################

Cons:

1. Choice of the loss functions: why the specific choice? Loss for misusing abstain class or mislabeling the true label could be something more general.

2. Choice of hyperparameters: How does one select \lamba_1 and \lambda_2 in practice? The authors mention a few practical ways (also used by authors in related works), e.g., ramp down of \kappa etc. Why are these good choices?

3. Clarity in the writing: The paper is unclear at many places- for example, it is nowhere mentioned that the maximum weighted class is finally chosen, and the interval bounds are not propagated at this last layer. Algorithm 1 is explained nowhere in the paper, except in a psuedocode.

4. Can similar benefits in adversarial robustness be obtained by post-processing an adversarially trained model?

##########################################################################

Questions during rebuttal period:


Please address and clarify the cons above


#########################################################################

---

> ### Author Response · Authors · 2020-11-14
> **Authors' Response Part 1/2**
>
> We thank the reviewer for their thoughtful comments. Let us start by first discussing the issue raised under “reason for score” in detail as it seems to be the major concern, and address the remaining comments next.
>
> Q: “... However, my major concern is that if these results could also be obtained  … by simply comparing "closeness" or confusion in the class labels ...”
>
> A: If we understand the suggestion correctly, it is proposed to define the detector such that an image is declared adversarial if the softmax output of the classifier corresponding to multiple classes are close (and both relatively high), or similarly, if the highest value of the softmax layer is relatively low. Although the proposed method is intuitively pleasing, the problem with such approaches lie in the assumption that the softmax output values of the classifier are indicative of how “certain” the classifier is in assigning different classes to the input image. However, it is not the case in practice, as it has been shown by various works in the literature that estimating the certainty of the network is nontrivial, and using similar metrics, e.g., the entropy of the softmax output by viewing it as a pmf, for detecting PGD-attacks fails as it yields ROC curves with AUC<<0.5; see “entropy- deterministic model ”curves in Fig 10 in [1]). This can be attributed to the objective of the PGD-attacks: its goal is to minimize the logit of the correct class, and maximize the logit of a wrong (or attack target) class. This objective thus creates adversarial examples for which almost all of the mass of the softmax output is put on the wrong class, thus rendering detectors based on “softmax-closeness” ineffective due to the seemingly (over)confidence of the classifier; see [1]. Overall, certainty estimation of neural networks is still an open area of research, and while other metrics, such as confidence calibration [2] and uncertainty estimation using Bayesian Networks [1], are proposed for increasing robustness, none yet have provable guarantees for adversarial example detection.
>
> More importantly however, the most elusive aspect of assessing any detector has proven to be testing them against adaptive attacks, as most detection methods have been shown ineffective once an adaptive attacker carefully identifies and targets their detection criteria, and successfully handles detection evasion while also causing misclassification. For a more detailed discussion, [3] provides a thorough analysis on how the majority of recent detection schemes are mostly ineffective against adaptive attacks.  Due to this factor, hand-crafting a detection criteria based on (mid or) final features of the classification network can often be ineffective as it can be explicitly targeted by carefully designed adaptive attacks.
>
> We hope this clarifies why we have approached the problem by refraining from using hand-crafted detection criteria, focusing on certifiability, and providing guarantees such that no (adaptive) attack can violate. To the best of our knowledge, this is the first work that can provide such guarantees.
>
>
> Q: “1. Choice of the loss functions: why the specific choice? Loss for misusing abstain class or mislabeling the true label could be  more general.”
>
> A: We are not certain which of the loss functions this comment is referring to, as there are various loss definitions such as Eq. (12), training loss in Eq. (13) and (14), or the adaptive attack loss in Eq. (36) in Appendix.  All these loss functions however, are based on Eq. (12), which is a general and direct result of the definition of attack failure in a joint classification/detection: “An attack is considered unsuccessful if the adversarially perturbed image is classified as either (a) the correct class, or (b) the abstain class.” Since the final outcome of the classifier is the class with the highest softmax-value, this forms the attacker’s objective to maximize the cross-entropy loss of the minimum of the two terms (correct-class or abstain-class cross-entropy losses) as defined in Eq. (12)
>
> The training loss in Eq. (13) is inspired by Eq. (12), but is slightly modified to accommodate its upperbounding during the training process, and Eq. (14) is motivated by empirical tests to help with stabilizing the training process, similar to that in [4]. Please note that no such modification (as in Eq. 13) is used during verification to insure exact certification. Additionally, the adaptive attack loss in Appendix is also based on the same definition of (un)successful attacks, however it is carried out in the logit-domain to ensure circumventing gradient obfuscation for better optimization via the PGD-solver. We hope the above comment is addressed, and please let us know if further clarification on any of these choices is needed.

---

> > ### Author Response · Authors · 2020-11-14
> > **Authors' Response Part 2/2**
> >
> > Q: “2. Choice of hyperparameters”
> >
> > A: The choice of parameters $\lambda_1$ and $\lambda_2$ corresponds to different levels of trade-off between the natural and robust errors, and thus tuning these parameters has indeed a strong dependence on the application and how this tradeoff is reflected in the desired outcome, e.g., how much the application field leans towards "correctly-classifying/detecting adversarial images" rather than "correctly classifying natural images". Values of $lambda_1$ and $lambda_2$ in Table 1 of the manuscript are selected such that natural-error is comparable with SOTA methods for a fair comparison, however, a more thorough study of their effect is also investigated in Subsection 5.3, in which the network is trained for various values of $\lambda_1$ and $\lambda_2$ and a wider range of the curve is plotted for comparison in Fig. 2 and 3. It is also interesting to note that the proposed algorithm has a better ability to trade off the natural and robust error by varying $\lambda_2$ in comparison with classification-only IBP. Parameter $\lambda_1$ on the other hand has a more subtle effect in trading off the abstain capability with robust-classification.
> >
> >
> > Q: “3. Clarity in the writing”
> >
> > A: Thank you for the suggestions. We have further clarified these points, added a step-by-step description for Algorithm 1 in the Appendix A.3 (due to space limitation), and will be revising the manuscript to address other potential ambiguities as well.
> >
> >
> >
> > Q: “4. Can similar benefits in adversarial robustness be obtained by post-processing...?”
> >
> > A: As discussed in the first part of our response in detail, using hand-crafted post-processing criteria for detection, despite its intuitive explanation, has shown to be problematic due to the fact that (a) network uncertainty is not necessarily reflected in the softmax outputs, e.g. a result of the PGD-attack loss function definition among other factors, and (b) adaptive attacks can often successfully evade detection by crafting perturbations that jointly attack such explicit detection criteria as well as cause misclassification. The same augment holds for adversarially trained networks as well; please see above for a more detailed discussion and references.
> >
> >
> > We hope that the raised concerns have been fully addressed, and we would be grateful to discuss in more details if any of the raised points remain unclear. We look forward to receiving your feedback and comments.
> >
> >
> >
> >
> > [1] Smith, L. and Gal, Y., 2018. Understanding measures of uncertainty for adversarial example detection. arXiv preprint arXiv:1803.08533.
> >
> > [2] Stutz, David, Matthias Hein, and Bernt Schiele. Confidence-calibrated adversarial training: Generalizing to unseen attacks. International Conference on Machine Learning. 2020.
> >
> > [3] Tramer, F., Carlini, N., Brendel, W. and Madry, A., 2020. On adaptive attacks to adversarial example defenses. arXiv preprint arXiv:2002.08347.
> >
> > [4] Gowal, Sven, et al. On the effectiveness of interval bound propagation for training verifiably robust models. arXiv preprint arXiv:1810.12715.

---

### Author Response · Authors · 2020-11-25
**Summary of Updates**

We would like to thank the reviewers for their comments and time in thoroughly reviewing our submission.
A revised version of the manuscript has been updated in which, together with the posted responses, we have carefully addressed the raised comments. The main discussed and updated points can be summarized as follows.

* Detection criteria and certifiability: By referencing previous works in the literature, we have addressed why “closeness of the classifier output” and other similar criteria cannot be utilized against adaptive attacks, justifying our deliberate refraining from pre-specified detection criteria. This, together with the fact that previous (non-certifiable) detection schemes have exhibited weak performance when tested against adaptive attacks- surprisingly sometimes broken prior to their publication dates; see [Athalye et al. ’18] - clarify the importance of certifiability of the proposed method, which to the best of our knowledge, is the first in the literature.

* Natural-vs-robust performance tradeoff and its optimization for joint detection/classification: This tradeoff is now extensively addressed in the updated manuscript, where subsection 5.3 under experiments is devoted to plotting the corresponding Pareto Frontiers for MNIST and CIFAR datasets for different values of $\epsilon$, further clarifying the improved curve when compared to classification without detection. As pointed by the reviewers, defending against adversarial examples comes at the price of reduced natural accuracy, and in this work we have aimed at showing that such trade-off can be improved by introducing a certifiable detection capability. In order to make sure we are providing a fair comparison, and the improvement is only due to the detection capability, we have fixed the bound propagation methodology to IBP for classification with and without detection, thus ensuring the improvement is not due to a possibly tighter/more-expensive relaxation scheme. Extending detection to more-realistic out-of-distribution samples rather than adversarial examples as well as optimizing the relaxation technique for tighter detection-enabled bounds are certainly among interesting future directions, and we believe the current work helps in paving the way towards these goals.

* Fair comparison: Given the unavailability of prior work on certifiable detection-based defenses together with the reduced performance of available (non-certifiable) detectors against adaptive attacks, and similar to some detection-based prior works, we have compared the performance of the proposed joint classification/detection with verifiably robust classification methods. Additionally, under “Effectiveness of the detection class” in subsection 5.2, discussion is provided to clarify the role of the detection class in the performance of classification with the detection capability and to differentiate the performance metric of the classifiers with and without detection, and Figure 1 also reports the percentage of the error/verification in natural/robust performance metrics that are attributed to the abstain class in tests on CIFAR-10 dataset.



Once again, we would like to thank the reviewers for their insightful comments. We hope the discussions and the updated manuscript have addressed and clarified the raised comments, and we would be grateful if the reviewers would consider the updated manuscript in their final reviews and decisions.

---

### Decision · Program_Chairs · 2021-01-07
**Final Decision**

**Decision:**

Accept (Poster)

**Comment:**

The paper studies the problem of certified adversarial robustness when the classifier has a reject option (realized here as an additional class) where the certification is done by adapting IBP-techniques to this particular problem setting.

Pro: As previous approaches to adversarial robustness with reject option have often be shown to be non-robust as one could circumvent both detector and classifier simultaneously with an adaptive attack, the idea to do this instead with a certified approach is interesting and could potentially initiate more research in this direction.

Con:
- The approach seems to some extent trade-off robust error with normal error which seems in particular true for MNIST where a stronger loss in normal error is less acceptable.

Comments:
- One reviewer mentioned that such a certified approach could be interesting for OOD detection. This has been done recently in:
Julian Bitterwolf, Alexander Meinke, Matthias Hein
Certifiably Adversarially Robust Detection of Out-of-Distribution Data, NeurIPS 2020
which should be cited in the present paper.
- While the empirical PGD-attacks done in this paper are strong, sometimes PGD fails due to gradient obfuscation, I would thus recommend to use additionally a black box attack or the recent AutoAttack.